# CAUSAL DISCOVERY VIA QUANTILE PARTIAL EFFECT

**Yikang Chen, Xingzhe Sun, Dehui Du**[*]
Shanghai Key Laboratory of Trustworthy Computing, East China Normal University

## ABSTRACT

Quantile Partial Effect (QPE) is a statistic associated with conditional quantile regression, measuring the effect of covariates at different levels. Our theory demonstrates that when the QPE of cause on effect is assumed to lie in a finite linear span, cause and effect are identifiable from their observational distribution. This generalizes previous identifiability results based on Functional Causal Models (FCMs) with additive, heteroscedastic noise, etc. Meanwhile, since QPE resides entirely at the observational level, this parametric assumption does not require considering mechanisms, noise, or even the Markov assumption, but rather directly utilizes the asymmetry of shape characteristics in the observational distribution. By performing basis function tests on the estimated QPE, causal directions can be distinguished, which is empirically shown to be effective in experiments on a large number of bivariate causal discovery datasets. For multivariate causal discovery, leveraging the close connection between QPE and score functions, we find that Fisher Information is sufficient as a statistical measure to determine causal order when assumptions are made about the second moment of QPE. We validate the feasibility of using Fisher Information to identify causal order on multiple synthetic and real-world multivariate causal discovery datasets.

## 1 INTRODUCTION

Multivariate causal discovery aims to elucidate inter-variable structures, yielding causal graphs or causal orders crucial for non-parametric identification and effect estimation in causal inference (Pearl, 2009). The sparsity of these graph structures aids downstream tasks such as feature selection (Guyon et al., 2007) and disentangled representation learning (Yang et al., 2021). To identify hidden Directed Acyclic Graphs (DAGs), various causal discovery methods have been developed, including constraint-based (Spirtes & Glymour, 1991; Spirtes, 1995) and score-based (Cooper & Herskovits, 1992; Chickering, 2002) approaches. However, these methods typically identify only up to an equivalence class, failing to distinguish causes from effects without additional assumptions.

Functional Causal Models (FCMs) are a class of Structural Causal Models (SCMs) that impose constraints on causal mechanisms. These include Additive Noise Models (ANM) (Hoyer et al., 2008), Heteroscedastic Noise Models (HNM) (Tagasovska et al., 2020), and Post-Nonlinear (PNL) models (Zhang & Hyvärinen, 2009). The inherent asymmetries in FCMs ensure that models satisfying these assumptions can distinguish cause from effect, except in some marginal cases. However, FCMs require strong assumptions about the underlying mechanism, noise, and Markov property, which may not hold in real-world scenarios. Recently, causal velocity (Xi et al., 2025) has generalized FCMs without requiring assumptions on functional form and noise. Nevertheless, it relies on counterfactual concepts, making it challenging to test the validity of the underlying counterfactual assumptions.

Inspired by causal velocity, this work discovers a more fundamental concept called Quantile Partial Effect (QPE), which reflects the shape characteristics of the observational distribution. This is a statistic purely at the observational level, defined by the conditional quantile function (Koenker, 2005) (Definition 3.1). It includes two equivalent definitions (Propositions 3.2 and 3.3). We also find cause-effect identifiability presents if QPE is assumed to lie in a finite linear span when basis functions are given (Assumption 3.5). This assumption generalizes past FCM assumptions (detailed in Table 1). In contrast to previous methods, however, identifiability via QPE relies purely on the observational distribution, independent of the underlying mechanism, noise, or Markov properties

---

[*]Corresponding author: dhdu@sei.ecnu.edu.cn

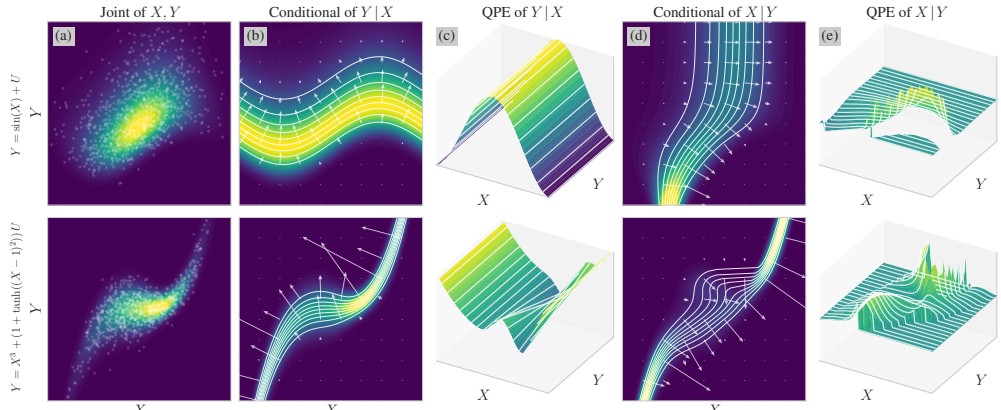

Figure 1: Distributions and their QPEs for ANM $Y = \sin(X) + U$ and HNM $Y = X^3 + (1 + \tanh((X - 1)^2)) U$. **(a)** Joint distribution (heatmap) and samples (scatterplot); **(b)** Conditional density of $Y \mid X$ (heatmap), conditional quantiles (white curves), and their gradients (white arrows); **(c)** QPE of $Y \mid X$ (3D surface) and its intersection with the Y-Z plane (white curves); **(d)** Conditional density and conditional quantiles of $X \mid Y$; **(e)** QPE of $X \mid Y$ (3D surface) and its intersection with the X-Z plane. ANM guarantees that the intersection of the QPE of $Y \mid X$ in the Y-Z plane is a constant function, while HNM guarantees it is an affine function, due to restrictions on the QPE form (see Table 1); the converse generally does not hold (Section 3.2).

(Corollary 3.8). Two algorithms for bivariate causal discovery based on QPE estimation and basis testing were subsequently developed (Sections 4.2 and 4.3). Experiments on bivariate causal discovery datasets demonstrate the effectiveness of these algorithms (Section 6.1).

When discussing multivariate causal discovery, we turn to indirect statistical measures to determine causal order, as estimating QPE becomes increasingly difficult in high dimensions. Given the close relationship between QPE and the score function (Lemma 3.4), we find that the second moment of QPE influences Fisher information (Theorem 5.2). Under certain assumptions regarding only QPE, Fisher information can sufficiently distinguish between cause and effect (Corollary 5.3). Based on this theory, we develop a simple and efficient non-parametric algorithm to identify causal orders, provided that Assumption 5.4 is satisfied. Finally, We validate the feasibility of this method on multiple synthetic and real-world multivariate causal discovery datasets (Section 6.2).

## 2 PRELIMINARIES

**Notation and General Assumptions**    Generally, this paper discusses random variables in $\mathbb{R}^d$. We denote a one-dimensional random variable and its realization as $X$ and $x$, respectively. For multi-dimensional random variables, we use $\boldsymbol{X} = (X_1, \ldots, X_d)$ and $\boldsymbol{x} = (x_1, \ldots, x_d)$. In some cases, lowercase boldfacing can also represent vector-valued functions, while uppercase boldfacing may denote matrices. Set operations between random variables refer to operations on their index sets and will also be used. For all probability density functions $p_{\boldsymbol{X}}$ appearing in this paper, we assume they are always existent, strictly positive, and at least $C^k$-functions when involving $k$-th derivatives.

**Structural Causal Models**    For a set of variables $\boldsymbol{X}$, an SCM is a triplet $(\boldsymbol{f}, \boldsymbol{X}, \boldsymbol{U})$, where $\boldsymbol{X}$ and $\boldsymbol{U}$ are endogenous and exogenous variables, respectively. For each $X_i \in \boldsymbol{X}$, we have $X_i = f_i(\boldsymbol{P}_i, U_i)$, where $\boldsymbol{P}_i \subseteq \boldsymbol{X} \backslash \{X_i\}$ are the parent variables and $f_i$ is the causal mechanism. An SCM is recursive if the graph $\mathcal{G}$ with nodes $\boldsymbol{X}$ and directed edges $(X_j, X_i)$ for every $X_j \in \boldsymbol{P}_i$ is a DAG. A causal order $\pi$ is any total order consistent with the topological sort of $\mathcal{G}$. A recursive SCM satisfies the Markov assumption if the exogenous variables are jointly independent (i.e., causal sufficiency or no latent confounders (Pearl, 2009)), which implies $\boldsymbol{P}_i \perp\!\!\!\perp U_i$ for each $X_i$.

**Score Function and Stein's Identity**    This paper refers to the gradient of the logarithmic density, $\nabla_{\boldsymbol{x}} \log p_{\boldsymbol{X}}$, as the score function. When a mild boundary condition, $\lim_{\boldsymbol{x} \to \pm\infty} \boldsymbol{h}(\boldsymbol{x}) \, p_{\boldsymbol{X}}(\boldsymbol{x}) = 0$,

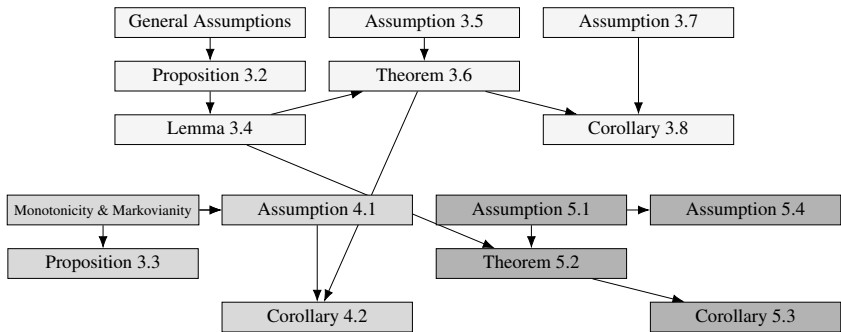

Figure 2: For clarity, we present a dependency graph of the assumptions and theorems in the main text, where "A→B" indicates that B depends on A: **(a)** Theorems in □ correspond to Section 3; **(b)** Theorems in □ correspond to Section 4; **(c)** Theorems in ▨ correspond to Section 5.

holds for a class of functions $\boldsymbol{h}$, we have Stein's Identity (Stein, 1972):

$$\mathbb{E}\left[\boldsymbol{h}(\boldsymbol{x})^{\mathsf{T}}\nabla_{\boldsymbol{x}}\log p_{\boldsymbol{X}}(\boldsymbol{x}) + \nabla_{\boldsymbol{x}}\cdot\boldsymbol{h}(\boldsymbol{x})\right] = 0.$$

A direct consequence is that the score function has zero expectation. Its variance, which is equivalent to its second moment, constitutes the Fisher Information. For a single variable $X_i$, the Fisher Information is the variance of the partial score $\partial_{x_i}\log p_{\boldsymbol{X}}$. Under the same regularity conditions, it equals the negative expected second derivative: $\mathbb{E}[(\partial_{x_i}\log p_{\boldsymbol{X}})^2] = -\mathbb{E}\left[\partial_{x_i}^2\log p_{\boldsymbol{X}}\right]$, where the second term is the negative expectation of a diagonal entry of the Hessian matrix of the log-density.

**Causal Normalizing Flows**   Javaloy et al. (2023) summarized the connection between autoregressive normalizing flows and SCMs, proposing to use flows to model SCMs and fit observational distributions, and showing their capacity for counterfactual inference. Taking a single causal mechanism $X_i = f_i(\boldsymbol{P}_i, U_i)$ as an example, assuming it is strictly monotonic w.r.t. $U_i$ and the SCM satisfies the Markov assumption, then according to the change-of-variable formula:

$$\log p_{X_i|\boldsymbol{P}_i} = \log p_{U_\theta} + \log|\partial_{x_i}u_\theta|, \tag{1}$$

where the causal flow is modeled as $u_\theta(\boldsymbol{p}_i, x_i)$, which maps the endogenous variable $X_i$ to a latent variable $U_\theta$, and $p_{U_\theta}$ is called the latent distribution. This formula also allows us to use Maximum Likelihood Estimation (MLE) to estimate parameters $\theta$ by maximizing $\mathbb{E}[\log p_{U_\theta} + \log|\partial_{x_i}u_\theta|]$. Javaloy et al. (2023) also discussed how to use autoregressive flows to fit multivariate SCMs and proved that under the above assumptions, such modeled causal flows ultimately lead to latent variables identifiable up to element-wise invertible transformations. Chen & Du (2025) then proved that causal flows and the true SCM are counterfactually consistent under these assumptions.

## 3   QPE FOR CAUSAL DISCOVERY

### 3.1   DEFINITION OF QPE AND EQUIVALENT CONCEPTS

We use QPE, a statistical object induced from conditional quantile regression, for causal discovery. QPE describes the sensitivity of quantiles to covariates, quantifying covariate effects at different levels. Its visualization, as shown in Figure 1, reflects the shape characteristics of the observational distribution. Let $F_{Y|\boldsymbol{X}}$ be the conditional cumulative distribution function (CDF) corresponding to the conditional distribution $p_{Y|\boldsymbol{X}}$, and $Q_{Y|\boldsymbol{X}}$ be the conditional quantile function. Then:

**Definition 3.1** (Quantile Partial Effect). The Quantile Partial Effect (QPE) of a random variable $Y$ given $\boldsymbol{X}$ is $\boldsymbol{\psi}_{Y|\boldsymbol{X}}(y|\boldsymbol{x}) = \nabla_{\boldsymbol{x}}Q_{Y|\boldsymbol{X}}(\tau|\boldsymbol{x})$, where the quantile $\tau = Q_{Y|\boldsymbol{X}}^{-1} = F_{Y|\boldsymbol{X}}$.

Based on the equality $F_{Y|\boldsymbol{X}} = Q_{Y|\boldsymbol{X}}^{-1}$, we can immediately find an equivalent description of QPE:

**Proposition 3.2** (QPE from CDF). $\boldsymbol{\psi}_{Y|\boldsymbol{X}} = -\nabla_{\boldsymbol{x}}F_{Y|\boldsymbol{X}}/\partial_y F_{Y|\boldsymbol{X}} = -\nabla_{\boldsymbol{x}}F_{Y|\boldsymbol{X}}/p_{Y|\boldsymbol{X}}.$

Xi et al. (2025) introduced the concept of causal velocity to generalize FCM-based bivariate causal discovery. Consider a Markovian causal mechanism $Y = f(\boldsymbol{X}, U)$, where $f$ is strictly monotonic

Table 1: The functional forms of FCMs and their corresponding QPE forms. For HNM, $b > 0$, and for PNL, $\overline{g} = g'(g^{-1})$. The QPEs of these FCMs can all be expressed in finite-rank forms.

| FCM | Functional Form | QPE Form | QPE Basis Functions |
|---|---|---|---|
| LiNGAM | $Y = \boldsymbol{c}^\mathsf{T}\boldsymbol{x} + u$ | $\boldsymbol{c}$ | 1 |
| ANM | $Y = a(\boldsymbol{x}) + u$ | $\nabla a$ | 1 |
| HNM | $Y = a(\boldsymbol{x}) + b(\boldsymbol{x})\,u$ | $\left(\nabla a - \frac{a}{b}\nabla b\right) + \frac{\nabla b}{b}\,y$ | $1, y$ |
| PNL-ANM | $Y = g(a(\boldsymbol{x}) + u)$ | $\nabla a\,\overline{g}$ | $\overline{g}$ |
| PNL-HNM | $Y = g(a(\boldsymbol{x}) + b(\boldsymbol{x})\,u)$ | $\left(\nabla a - \frac{a}{b}\nabla b\right)\overline{g} + \frac{\nabla b}{b}\,g^{-1}\,\overline{g}$ | $\overline{g}, g^{-1}\,\overline{g}$ |
| Assumption 3.5 | Perhaps no closed-form | $\sum_{j=1}^{k} c_j(\boldsymbol{x})\,\phi_j(y)$ | $\boldsymbol{\phi}$ |

w.r.t. $U$. The term $f(\boldsymbol{x}', u(\boldsymbol{x}, y))$ is referred to as the counterfactual outcome (or SCM flow), where $u = (f(\boldsymbol{x}, \cdot))^{-1}$. Causal velocity is then defined as $\nabla_{\boldsymbol{x}} f(\boldsymbol{x}, u)$. According to (Nasr-Esfahany et al., 2023), under these assumptions, the counterfactual outcome is identifiable from the observational distribution, meaning causal velocity is independent of the exogenous distribution. In fact, we find

**Proposition 3.3** (Causal Velocity is QPE). $\boldsymbol{\psi}_{Y|\boldsymbol{X}} = \nabla_{\boldsymbol{x}} f(\boldsymbol{x}, u) = -\nabla_{\boldsymbol{x}} u / \partial_y u$, a.e.

According to the Causal Hierarchy Theory (Bareinboim et al., 2022), this implies that causal velocity, a counterfactual quantity, can "collapse" entirely into an observational quantity, despite its definition through counterfactual concepts. In contrast to causal velocity, QPE does not require monotonic mechanisms or the Markov assumption. Its definition (Definition 3.1) and cause-effect identifiability (Section 3.2) depend solely on the observational distribution.

Xi et al. (2025) also found a PDE relationship between causal velocity and the score function, based on the SCM flow and the continuity equation. This PDE naturally applies to QPE due to their equivalence. In fact, by directly taking the partial derivative of the implicit function defined in Proposition 3.2, we can derive this equation:

**Lemma 3.4.** $\nabla_{\boldsymbol{x}} \log p_{Y|\boldsymbol{X}} + \boldsymbol{\xi}\,\partial_y \log p_{Y|\boldsymbol{X}} + \partial_y \boldsymbol{\xi} = 0$ and $\lim_{y \to \pm\infty} \boldsymbol{\xi}\, p_{Y|\boldsymbol{X}} = 0$ iff $\boldsymbol{\xi} = \boldsymbol{\psi}_{Y|\boldsymbol{X}}$.

Now consider each covariate $X_i$, and denote the corresponding component in QPE as $\psi_{Y|\boldsymbol{X},i} = \partial_{x_i} Q_{Y|\boldsymbol{X}}$. Since $\partial_{x_i} \log p_{\boldsymbol{X}}$ is independent of $Y$, according to $\log p_{\boldsymbol{X},Y} = \log p_{\boldsymbol{X}} + \log p_{Y|\boldsymbol{X}}$, we can obtain the equality $\partial_y \log p_{Y|\boldsymbol{X}} = \partial_y \log p_{\boldsymbol{X},Y}$ between the conditional log-density and the joint log-density. Therefore, a second-order mixed PDE can be derived from Lemma 3.4:

$$\partial_{x_i}\partial_y \log p_{\boldsymbol{X},Y} + \partial_y\big(\psi_{Y|\boldsymbol{X},i}\,\partial_y \log p_{\boldsymbol{X},Y} + \partial_y \psi_{Y|\boldsymbol{X},i}\big) = 0, \tag{2}$$

which only involves the score function of the joint distribution and QPE.

## 3.2 Cause-Effect Identifiability by QPE in Finite Linear Span

For a function $f : \mathbb{R} \to \mathbb{R}$, if it can be represented as a linear combination of a set of basis functions $\boldsymbol{\phi} = (\phi_1, \ldots, \phi_k)$ such that there exist coefficients $c_i, \ldots, c_k$ satisfying $f(x) = \sum_{j=1}^{k} c_j\,\phi_j(x)$, then $f$ is said to be in the finite linear space spanned by $\boldsymbol{\phi}$, denoted as $f \in \mathrm{span}(\boldsymbol{\phi})$.

**Assumption 3.5.** For each cause variable $X_i$ and any $\boldsymbol{x}$, $\psi_{Y|\boldsymbol{X},i}(\cdot \mid \boldsymbol{x}) \in \mathrm{span}(\boldsymbol{\phi})$, where $\boldsymbol{\phi}$ is a known set of basis functions that depend only on the effect variable $Y$.

In other words, this assumption requires that each component $\psi_{Y|\boldsymbol{X},i}(\cdot \mid \boldsymbol{x})$ can be represented in a finite-rank form as a sum of finite $c_{i,j}(\boldsymbol{x})\,\phi_j(y)$, where $c_{i,j}$ are coefficient functions w.r.t. $\boldsymbol{x}$.

We observe that previous FCMs with constrained functional forms can in fact be expressed in a finite-rank form, as detailed in Table 1. However, note that Assumption 3.5 requires a known $\boldsymbol{\phi}$; thus, although it generalizes ANM and HNM, it does not directly apply to PNL where $g$ is unknown. Furthermore, this assumption is entirely independent of monotonicity or the Markov property, distinguishing our approach from the generalization path taken by Xi et al. (2025). Specific examples that violate these properties yet still satisfy Assumption 3.5 are provided in Appendix A.3.

Next, we consider constructing equations solely dependent on the joint distribution using linear relationships. For a set of multivariate functions $f_1, \ldots, f_k$, where each $f_i : \mathbb{R}^d \to \mathbb{R}$, the determinant

$$W_X(f_1, \ldots, f_k) = \det \begin{bmatrix} f_1 & f_2 & \cdots & f_k \\ \partial_x f_1 & \partial_x f_2 & \cdots & \partial_x f_k \\ \vdots & \vdots & \ddots & \vdots \\ \partial_x^{k-1} f_1 & \partial_x^{k-1} f_2 & \cdots & \partial_x^{k-1} f_k \end{bmatrix}$$

is called the Wronskian determinant w.r.t. the variable $X$. Let $s_{X_i,Y} = \partial_{x_i} \partial_y \log p_{\boldsymbol{X},Y}$ and $\boldsymbol{\eta}_{Y|\boldsymbol{X}} = \partial_y \big( \boldsymbol{\phi} \, \partial_y \log p_{\boldsymbol{X},Y} + \boldsymbol{\phi}' \big)$. Here, $s_{X_i,Y}$ is an off-diagonal element in the Hessian matrix, while $\boldsymbol{\eta}_{Y|\boldsymbol{X}}$ is the partial derivative of $\boldsymbol{\phi}$'s Stein operator w.r.t. $y$. Then, according to the second-order mixed PDE (Equation 2), the following theorem holds:

**Theorem 3.6** (Identifiability of QPE in Finite Linear Span). *For each variable $X_i$ and any $\boldsymbol{x}$, $\psi_{Y|\boldsymbol{X},i}(\cdot \mid \boldsymbol{x}) \in span(\boldsymbol{\phi})$ implies $W_Y(s_{X_i,Y}, \boldsymbol{\eta}_{Y|\boldsymbol{X}}) = 0$ for any $y$. If: (i) The components in $\boldsymbol{\eta}_{Y|\boldsymbol{X}}$ are linearly independent; (ii) There exists $y$ such that $W_Y(\boldsymbol{\eta}_{Y|\boldsymbol{X}}) \neq 0$; (iii) For each basis function $\phi_j$, $\lim_{y \to \pm\infty} \phi_j \, p_{Y|\boldsymbol{X}} = 0$; then the converse is also true.*

Theorem 3.6 constitutes a necessary (or sufficient, assuming well-behaved conditions) condition for the validity of Assumption 3.5. Crucially, Theorem 3.6 relies solely on the Wronskian determinant, which is a function only of the joint distribution and the basis set $\boldsymbol{\phi}$ (as well as their higher-order derivatives). By leveraging this Wronskian determinant, we can formally characterize the asymmetry of shape characteristics in the observational distribution given $\boldsymbol{\phi}$, and further achieve cause-effect identifiability that relates only to the observational distribution:

**Assumption 3.7.** The set of basis functions $\boldsymbol{\phi}$ is known, and for any $Z \in \boldsymbol{X} \setminus Y$ and any $X_i \in \boldsymbol{X} \setminus Z$, we have $W_Z(s_{X_i,Z}, \boldsymbol{\eta}_{Z|\boldsymbol{X}}) \neq 0$.

**Corollary 3.8** (Cause-Effect Identifiability by QPE in Finite Linear Span). *If Assumption 3.5 and Assumption 3.7 hold simultaneously, then $Y$ is the effect variable.*

**Proof** Theorem 3.6 establishes a necessary condition for Assumption 3.5 ($\psi_{Y|\boldsymbol{X},i} \in span(\boldsymbol{\phi})$) to hold is $W_Z(s_{X_i,Y}, \boldsymbol{\eta}_{Y|\boldsymbol{X}}) = 0$, so Assumption 3.7 implies $\psi_{Z|\boldsymbol{X},i} \notin span(\boldsymbol{\phi})$. Consequently, Assumption 3.5 does not hold for any $Z \in \boldsymbol{X} \setminus Y$, leaving $Y$ as the only possible effect variable.

It is worth noting that both Assumption 3.5 and Assumption 3.7, as utilized in Corollary 3.8, depend solely on the observational distribution (assuming $\boldsymbol{\phi}$ is pre-specified). The former is independent of monotonic mechanisms and the Markov assumption, as illustrated in Appendix A.3; the latter describes a PDE system over the observational distribution, reflecting its asymmetry. Consequently, via QPE, we achieve cause-effect identifiability based strictly on the observational distribution.

## 4 QPE FOR BIVARIATE CAUSAL DISCOVERY

### 4.1 QUANTIFY CAUSE-EFFECT IDENTIFIABILITY THROUGH THE LENS OF FCM

While Corollary 3.8 establishes cause-effect identifiability through Assumption 3.5 and Assumption 3.7, weaknesses remain. Most notably, characterizing the probability that Assumption 3.7 holds is challenging. To resolve this, we revert to the bivariate FCM framework, replacing Assumption 3.7 with the assumptions of monotonic mechanisms and the Markov property. This enables a quantitative description of cause-effect unidentifiability grounded in manifold theory.

**Assumption 4.1.** A bivariate SCM $(f, X, U)$ such that $Y = f(X, U)$, where the causal mechanism $f$ and exogenous distribution $p_U$ are given. Furthermore, $f$ is strictly monotonic w.r.t. $U$, and the Markov property or $X \perp\!\!\!\perp U$ holds.

Assumption 4.1 describes a family of FCMs where only $p_X$ of $X$ is unspecified. All possible $p_X$ distributions form an infinite-dimensional manifold, $\boldsymbol{\Theta}$, denoted as $\dim(\boldsymbol{\Theta}) = \infty$, representing the scale of this FCM family. We then introduce forward and backward versions of Assumption 3.5 on QPE: "for any $x$, $\psi_{Y|X}(\cdot \mid x) \in span(\boldsymbol{\phi})$" or "for any $y$, $\psi_{X|Y}(\cdot \mid y) \in span(\boldsymbol{\phi})$". These conditions induce two submanifolds, $\boldsymbol{\Theta}_{X \to Y;\phi}$ and $\boldsymbol{\Theta}_{X \leftarrow Y;\phi}$, respectively. These submanifolds contain all $p_X$ that satisfy the corresponding assumption and must at least satisfy Theorem 3.6.

**Corollary 4.2.** *Assume that there are $k$ basis functions in $\phi$ and certain regularity conditions hold. Then: (i) $\dim(\boldsymbol{\Theta}_{X \to Y;\phi}) = \infty$; (ii) $\dim(\boldsymbol{\Theta}_{X \leftarrow Y;\phi}) \leq k + 2$.*

In other words, under the premise of an FCM satisfying Assumption 4.1, if the forward version of Assumption 3.5 also holds, the choice for $p_X$ remains arbitrary. However, if the backward version of Assumption 3.5 holds, $p_X$ is constrained to lie within a submanifold of at most $k + 2$ dimensions. It is obvious then that the unidentifiable scenario, where both the forward and backward versions of the assumption hold simultaneously, also falls within such an at most $k + 2$-dimensional submanifold. Since this manifold is finite-dimensional, it is typically of measure zero for non-degenerate measures on an infinite-dimensional space. Therefore, generally, except for extremely marginal cases, Assumption 3.5 universally provides identifiability of causal direction.

While Corollary 4.2 retains the need for monotonic mechanisms and the Markov property, it offers guarantees that serves as a generalization of quantifiable cause-effect identifiability under the classic FCM framework. In particular, this corollary generalizes the conclusion by Hoyer et al. (2008) that unidentifiable ANMs lie in a three-dimensional affine space (since ANMs correspond to $k = 1$ where the Wronskian simplifies to a linear ODE) to any finite-linearly spanned QPE ($k < \infty$).

## 4.2 Kernel-based QPE with Least Square Basis Test

With the identifiability guarantees established above, we can determine the causal direction by estimating the QPE and checking whether it lies within the assumed span. The original definition of QPE suggests calculation via conditional quantile regression, but its accuracy and efficiency depend on the underlying quantile regression. Below is an efficient and entirely non-parametric method for bivariate causal discovery, independent of quantile estimation.

**Kernel-based QPE**  By Proposition 3.2, we can use non-parametric methods to estimate $\nabla_{\boldsymbol{x}} F_{Y|\boldsymbol{X}}$ and $\partial_y F_{Y|\boldsymbol{X}}$, thereby indirectly computing $\boldsymbol{\psi}_{Y|\boldsymbol{X}}$. Given $N$ samples $(\boldsymbol{x}_j, y_j)$ drawn from $p_{\boldsymbol{X},Y}$, the conditional CDF can be estimated using a kernel estimator as:

$$\hat{F}_{Y|\boldsymbol{X}}(y \mid \boldsymbol{x}) = \frac{\sum_{i=1}^{N} K(\boldsymbol{x}_i, \boldsymbol{x})\, S(y_i, y)}{\sum_{i=1}^{N} K(\boldsymbol{x}_i, \boldsymbol{x})},$$

where $K$ is a smoothed version of the indicator function $\mathbf{1}(\boldsymbol{x}_i \in \delta(\boldsymbol{x}))$ and $S$ is a smoothed version of the $\mathbf{1}(\boldsymbol{y}_i \leq y)$. We choose $K$ as a Gaussian kernel and $S$ as a sigmoid function. Based on this equation, since the kernel functions are known and smooth, we can derive closed-form expressions for $\nabla_{\boldsymbol{x}} \hat{F}_{Y|\boldsymbol{X}}(y \mid \boldsymbol{x})$ and $\partial_y \hat{F}_{Y|\boldsymbol{X}}(y \mid \boldsymbol{x})$.

**Least Square Basis Test**  After obtaining the estimated QPE $\hat{\boldsymbol{\psi}}_{Y|\boldsymbol{X}}$, we can test whether the estimate satisfies Assumption 3.5 at fixed $(\boldsymbol{x}, y)$ pairs. Specifically, we can pre-select $\{y_1, \ldots, y_M\}$ as test locations for $y$ and construct a basis matrix $\boldsymbol{B}$ such that $B_{m,j} = \phi_j(y_m)$. Concurrently, we select $\{\boldsymbol{x}_1, \ldots, \boldsymbol{x}_T\}$ as test samples for $\boldsymbol{x}$. For the $i$-th component, based on these fixed locations, we construct a response matrix $\boldsymbol{\Psi}_i$ such that $\Psi_{i,t,m} = \psi_{Y|\boldsymbol{X},i}(y_m \mid \boldsymbol{x}_t)$ for each $y_m$ and $\boldsymbol{x}_t$. If the true QPE $\psi_{Y|\boldsymbol{X},i}(y \mid \boldsymbol{x}) \in \mathrm{span}(\phi)$, then for a coefficient matrix $\boldsymbol{C}_i$ such that $C_{i,t,j} = c_{i,j}(\boldsymbol{x}_m)$, it must hold that $\boldsymbol{\Psi}_i = \boldsymbol{C}_i \boldsymbol{B}^{\mathsf{T}}$. Since $\boldsymbol{C}_i$ is unknown, this problem can be modeled as an Ordinary Least Squares (OLS) problem where the noisy $\hat{\boldsymbol{\Psi}}_i$ is treated as a linear response w.r.t. $\boldsymbol{B}$. Then,

$$\arg\min_{\hat{\boldsymbol{C}}_i} \left\| \hat{\boldsymbol{\Psi}}_i - \hat{\boldsymbol{C}}_i \boldsymbol{B}^{\mathsf{T}} \right\| \implies \hat{\boldsymbol{C}}_i = \hat{\boldsymbol{\Psi}}_i \boldsymbol{B} (\boldsymbol{B}^{\mathsf{T}} \boldsymbol{B})^{-1},$$

where, because the $\phi_j$ are not necessarily linearly independent across the $M$ test locations, which means that the inverse of $\boldsymbol{B}^{\mathsf{T}} \boldsymbol{B}$ may not exist, we use the pseudoinverse $(\boldsymbol{B}^{\mathsf{T}} \boldsymbol{B})^+$ instead to ensure numerical stability. Finally, we average the residuals for each component:

$$\varepsilon_{\boldsymbol{X} \to Y} = -\frac{1}{d} \sum_{i=1}^{d} \left\| \hat{\boldsymbol{\Psi}}_i - \hat{\boldsymbol{\Psi}}_i \boldsymbol{B} (\boldsymbol{B}^{\mathsf{T}} \boldsymbol{B})^+ \boldsymbol{B}^{\mathsf{T}} \right\|,$$

This value reflects the degree to which Assumption 3.5 is satisfied: $\varepsilon_{\boldsymbol{X} \to Y} = 0$ only if $\psi_{Y|\boldsymbol{X},i}(\cdot \mid \boldsymbol{x}) \in \mathrm{span}(\phi)$ holds for all covariates $X_i$. In bivariate causal discovery, we can identify the causal direction by comparing $\varepsilon_{X \to Y}$ and $\varepsilon_{Y \to X}$. We infer $Y$ as the effect if $\varepsilon_{X \to Y} > \varepsilon_{Y \to X}$, and $X$ as the effect otherwise.

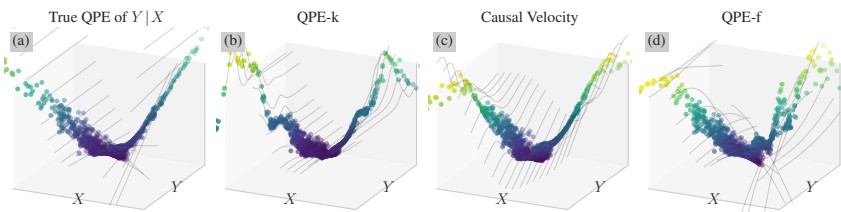

Figure 3: True and estimated QPEs of $Y \mid X$ at samples from HNM $Y = X^3 + (1 + \tanh((X - 1)^2)) U$. From left to right: **(a)** True QPE; **(b)** QPE-k (Section 4.2); **(c)** Causal velocity model (Xi et al., 2025) (V-NN); **(d)** QPE-f (Section 4.3). The black lines represent the intersection of the QPE surface with the Y-Z plane. Only QPE-f's trend tends to match the true QPE in high-density areas.

### 4.3 FLOW-BASED QPE WITH NEURAL BASIS TEST

While kernel methods combined with the least squares basis test offer a fast and effective non-parametric approach, their performance is limited by the choice of kernel bandwidth and sample size. QPE estimation tends to be less accurate and prone to overly smooth predictions. We will now introduce a neural network-based parametric method, which, despite requiring neural network training, often yields more accurate QPE estimates, as shown in Figure 3. In experiments, this method outperforms most state-of-the-art bivariate causal discovery methods.

**Flow-based QPE** By Proposition 3.3, for any SCM that satisfies the assumption, we can calculate the QPE using the function $u$ that maps the observational distribution to an exogenous distribution. According to Section 2, any causal flow modeled by Equation 1 is an SCM satisfying Proposition 3.3, and these SCMs are counterfactually identifiable. Thus, we can estimate the QPE through any flow $u_\theta$ parameterized as in Equation 1 and optimized via MLE. We use the standard normal distribution as the latent distribution and parameterize it with a neural network. After obtaining the causal flow $u_\theta$, we can compute $\nabla_{\boldsymbol{x}} u_\theta$ and $\partial_y u_\theta$ using automatic differentiation techniques, thereby calculating the QPE $\hat{\psi}_{Y|\boldsymbol{X}} = \nabla_{\boldsymbol{x}} u_\theta / \partial_y u_\theta$. Note that Proposition 3.3 implicitly guarantees that the QPE obtained this way is always consistent, even if a different latent distribution is chosen.

**Neural Basis Test** Although the previously described OLS-based test still applies to the estimated $\hat{\psi}_{Y|\boldsymbol{X}}$ here, it requires fixed $y$ values and cannot directly test based on irregularly distributed samples. Furthermore, OLS corresponds to MLE under the assumption that errors follow a Gaussian distribution. To address these issues, we consider using a parameterized neural network to directly model the coefficient function $c_{i,j,\theta}(\boldsymbol{x})$, which corresponds to optimizing the objective:

$$\arg \min_\theta \varepsilon_{\boldsymbol{X} \to Y, \theta} + \lambda \|\theta\|, \quad \text{where } \varepsilon_{\boldsymbol{X} \to Y, \theta} = \mathbb{E}\left[\left\|\hat{\psi}_{Y|\boldsymbol{X}} - \boldsymbol{C}_\theta \, \boldsymbol{\phi}^\mathsf{T}\right\|\right],$$

where $\lambda$ is a regularization hyperparameter, and $\boldsymbol{C}_\theta$ is the matrix formed by $c_{i,j,\theta}$. In contrast to the OLS test, here the samples $\boldsymbol{x}_t, y_m$ can be irregularly distributed. The optimal $\varepsilon_{\boldsymbol{X} \to Y, \theta}$ will serve as a measure of how well Assumption 3.5 is satisfied, and its use in determining the causal direction is the same as described previously.

This testing method can be simplified, as proposed by (Xi et al., 2025), to directly fit $\psi_{Y|\boldsymbol{X}}$ using unconstrained neural networks, called V-NN models. Specifically, they consider directly optimizing PDE in Lemma 3.4, which is guaranteed to converge under the vanishing assumption. However, Lemma 3.4 involves first-order gradients, and the performance heavily relies on the accuracy of score function estimation algorithm.

## 5 FISHER INFORMATION FROM THE QPE PERSPECTIVE FOR MULTIVARIATE CAUSAL DISCOVERY

### 5.1 THE CONNECTION BETWEEN FISHER INFORMATION AND QPE

Due to the curse of dimensionality, QPE estimation in high-dimensional settings performs poorly, limiting its direct application in multivariate causal discovery. Since Lemma 3.4 describes a de-

terministic relationship between the score function and QPE, we consider constructing an equation between the Fisher information and QPE. We then indirectly infer the Fisher information by assuming QPE, enabling its use for causal discovery. For variables $\boldsymbol{X}, Y$, we denote the score functions of the joint and marginal distributions, $\partial_{x_i} \log p_{\boldsymbol{X},Y}$ and $\partial_{x_i} \log p_{\boldsymbol{X}}$, as $s_{X_i}$ and $r_{X_i}$ respectively.

**Assumption 5.1.** Both $\psi_{Y|\boldsymbol{X},i}\, p_{\boldsymbol{X},Y}$ and $\psi_{Y|\boldsymbol{X},i}(\partial_y \psi_{Y|\boldsymbol{X},i})\, p_{\boldsymbol{X},Y}$ vanish at $\pm\infty$.

**Theorem 5.2.** *For each covariate $X_i$, if Assumption 5.1 holds then*

$$\mathbb{E}\left[(\psi_{Y|\boldsymbol{X},i})^2\,(s_Y)^2\right] = \mathbb{E}\left[(s_{X_i})^2\right] - \mathbb{E}\left[(r_{X_i})^2\right] + \mathbb{E}\left[(\partial_y \psi_{Y|\boldsymbol{X},i})^2 + 2\,\psi_{Y|\boldsymbol{X},i}\,\partial_y^2 \psi_{Y|\boldsymbol{X},i}\right].$$

**Corollary 5.3.** *For each covariate $X_i$, if Assumption 5.1 holds then*

$$\mathbb{E}\left[(\partial_y \psi_{Y|\boldsymbol{X},i})^2 + 2\,\psi_{Y|\boldsymbol{X},i}\,\partial_y^2 \psi_{Y|\boldsymbol{X},i}\right] < \mathbb{E}\left[((\psi_{Y|\boldsymbol{X},i})^2 - 1)\,(s_Y)^2\right] + \mathbb{E}\left[(r_{X_i})^2\right]$$

*if and only if $\mathbb{E}\left[(s_{X_i})^2\right] > \mathbb{E}\left[(s_Y)^2\right]$.*

This implies that if the marginal Fisher Information is sufficiently large, or the second moment of $\psi_{Y|\boldsymbol{X},i}$ is sufficiently large and the moments of the higher-order partial derivatives of $\psi_{Y|\boldsymbol{X},i}$ w.r.t. $y$ are sufficiently small, we can directly distinguish the effect variable by the Fisher Information.

**Qualitative analysis under the heteroscedastic Gaussian assumption** By introducing additional assumptions on the second-order partial derivative of $\psi_{Y|\boldsymbol{X},i}$ (i.e., HNM), we can isolate its second moment from the inequality in Corollary 5.3:

$$\mathbb{E}\left[(\psi_{Y|\boldsymbol{X},i})^2\right] > 1 - \frac{\mathbb{E}\left[(r_Y)^2\right]}{\mathbb{E}\left[(s_Y)^2\right]} + \frac{\mathbb{E}\left[(\partial_y \psi_{Y|\boldsymbol{X},i})^2\right] + \sqrt{\mathrm{Var}\left[(\psi_{Y|\boldsymbol{X},i})^2\right]\mathrm{Var}\left[(s_Y)^2)\right]}}{\mathbb{E}\left[(s_Y)^2\right]}.$$

Then, for Corollary 5.3 to be as valid as possible, we need to make assumptions about QPE: $\mathbb{E}[(\psi_{Y|\boldsymbol{X},i})^2]$ must be sufficiently large, and one of the following must hold: **(i)** $\mathrm{Var}[(\psi_{Y|\boldsymbol{X},i})^2]$ is sufficiently small when $\partial_y \psi_{Y|\boldsymbol{X},i} = 0$ (e.g., ANM); **(ii)** $\mathrm{Var}[(\psi_{Y|\boldsymbol{X},i})^2]$ is sufficiently small and $\mathbb{E}[(\partial_y \psi_{Y|\boldsymbol{X},i})^2]$ is sufficiently small when $\partial_y^2 \psi_{Y|\boldsymbol{X},i} = 0$ (e.g., HNM). Qualitatively, when assuming $\mathbb{E}[(\partial_y \psi_{Y|\boldsymbol{X},i})^2]$ is relatively stable, this trend requires the coefficient of variation (CV) of the squared QPE, $\sqrt{\mathrm{Var}[(\psi_{Y|\boldsymbol{X},i})^2]}/\mathbb{E}[(\psi_{Y|\boldsymbol{X},i})^2]$, to be sufficiently small.

Furthermore, if we assume a heteroscedastic Gaussian conditional distribution, i.e., $Y \mid X = x \sim \mathcal{N}(\mu(x), (\sigma(x))^2)$, we can derive the exact value of the CV of the QPE under linear $\mu, \sigma$, or its upper bound in the general case (see Appendix B.2 for the complete derivation):

$$\frac{\sqrt{\mathrm{Var}\left[(\psi_{Y|X})^2\right]}}{\mathbb{E}\left[(\psi_{Y|X})^2\right]} = \frac{\sqrt{4 + 2\kappa^2}}{1 + \kappa^2}|\kappa|, \qquad \frac{\sqrt{\mathrm{Var}\left[(\psi_{Y|X})^2\right]}}{\mathbb{E}\left[(\psi_{Y|X})^2\right]} \leq \left(\frac{\sqrt{1 + 6u^2 + 3u^4}}{1 + l^2}\right)\frac{\sqrt{\mathbb{E}\left[(\mu')^4\right]}}{\mathbb{E}\left[(\mu')^2\right]},$$

where $\kappa = \sigma'/\mu'$ and $l \leq |\kappa| \leq u$.

Based on the above analysis, qualitatively, under the heteroscedastic Gaussian assumption, the CV of QPE is sufficiently small when: **(i)** $|\kappa| = |\sigma'/\mu'|$ is sufficiently small, meaning $\mu$ rather than $\sigma$ dominates the shape of the distribution; and **(ii)** $\mu$ and $\sigma$ are close to linear functions. To further confirm the correctness of this qualitative analysis, we conducted experiments on datasets with widely varying average $|\kappa|$ and linearity. The results are presented in Appendix D.4.

## 5.2 Fisher Information Causal Ordering

**Assumption 5.4.** For each variable $X_i$, any of its parents $P_j \in \boldsymbol{P}_i$ satisfy Assumption 5.1 and

$$\mathbb{E}\left[(\partial_y \psi_{X_i|\boldsymbol{P}_i,j})^2 + 2\,\psi_{X_i|\boldsymbol{P}_i,j}\,\partial_y^2 \psi_{X_i|\boldsymbol{P}_i,j}\right] < \mathbb{E}\left[(\psi_{X_i|\boldsymbol{P}_i,j} - 1)^2\,(s_Y)^2\right] + \mathbb{E}\left[(r_{X_i})^2\right].$$

According to Corollary 5.3, if both Assumption 5.1 and Assumption 5.4 hold for a given set of variables $\boldsymbol{X}$, then the variable $X_i$ with the minimal Fisher information $\mathbb{E}[(s_{X_i})^2]$ must have no child variables, called the leaf variable. Therefore, Assumption 5.4 is a crucial simplification, implying that once the score function is estimated, there is a possibility of finding the leaf node. After selecting a leaf node $X_i$, a subproblem on $\boldsymbol{X}_{-i}$ is formed, which allows for recursion until all variables are removed. The

---

**Algorithm 1** Fisher Information Causal Ordering

**Input:** a set of $d$ random variables $\boldsymbol{X}$.
**Output:** causal order $\pi$
$\boldsymbol{X}^{(1)} \leftarrow \boldsymbol{X}, \pi \leftarrow [\,]$
**for** $j = 1$ **to** $d$ **do**
$\quad l \leftarrow \arg\min_i \mathbb{E}\left[\left(\partial_{x_i} \log p_{\boldsymbol{X}^{(j)}}\right)^2\right]$
$\quad \boldsymbol{X}^{(j+1)} \leftarrow \boldsymbol{X}^{(j)}_{-l}, \pi \leftarrow [l, \pi]$
**end for**
**return** $\pi$

---

resulting sequence of variables is the reverse of the causal order. This process is called **FICO** (Fisher Information Causal Ordering), and the detail is described in Algorithm 1.

Next, given the causal order, for $d$ variables, we only need to perform at most $d(d-1)/2$ conditional independence tests to prune the full DAG induced by this topological order, thereby retaining only the necessary edges. Under the assumptions of causal sufficiency and faithfulness, independence tests ensure that the DAG lies within the Markov equivalence class, and the direction of each edge is determined by Corollary 5.3. Thus, the resulting DAG is fully identifiable.

Notably, CaPS (Xu et al., 2024), another score function based causal discovery method, is algorithmically fully equivalent to FICO. However, distinct differences exist between them: **(i)** CaPS is specifically tailored for the ANM, and its theoretical derivation relies on the ANM assumption. In contrast, FICO applies to any model, provided that the vanishing assumption (Assumption 5.1) holds and the QPE satisfies Assumption 5.4. **(ii)** CaPS utilizes $-\mathbb{E}\left[\partial_{x_i}^2 \log p_{\boldsymbol{X}}\right]$, whereas FICO employs $\mathbb{E}\left[(\partial_{x_i} \log p_{\boldsymbol{X}})^2\right]$. Although both are equivalent, FICO involves lower-order derivatives, resulting in higher computational efficiency (see Section 6.2 for details).

It is important to emphasize that although FICO's assumptions and theory have been extended to general cases, and despite its simplicity and computational efficiency, it still faces challenges worth addressing: **(i)** The assumption relied upon by FICO (Assumption 5.4) has only been analyzed qualitatively under the heteroscedastic Gaussian setting. We still lack a precise understanding of what the validity of Assumption 5.4 implies in broader contexts. **(ii)** Calculating the QPE in high-dimensional settings is difficult, which renders the testability of Assumption 5.4 problematic. Given these limitations, caution is still advised when applying FICO in practice.

## 6 EXPERIMENTS

### 6.1 BIVARIATE CAUSAL DISCOVERY EXPERIMENTS

**Datasets** We conducted experiments on 24 synthetic and real-world benchmarks, including: **(i)** AN, AN-c, LS, LS-c, MNU from (Tagasovska et al., 2020); **(ii)** SIM, SIM-c, SIM-g, SIM-ln from (Mooij et al., 2016); **(iii)** Cha, Multi, Net from (Guyon et al., 2019); **(iv)** Per, Sig, Vex generated by random SCM flows from (Xi et al., 2025); **(v)** Qd-V, Sig-V, Rbf-V, NN-V generated by constrained QPEs (Appendix D.2); **(vi)** Tübingen cause-effect pairs challenge (Mooij et al., 2016); **(vii)** Gene network reverse engineering challenge from (Marbach et al., 2009). Datasets **(vi)** and **(vii)** are real-world datasets, while the underlying SCMs of **(iv)** and **(v)** are not necessarily ANM, HNM or PNL.

**Baselines** We consider the following open source methods as baselines: **(i)** ANM and PNL with independence test, implemented by (Zheng et al., 2024); **(ii)** ANM-based RECI (Bloebaum et al., 2018) and CDS (Fonollosa, 2019), with implementations from (Kalainathan et al., 2020); **(iii)** HNM-based CDCI (Duong & Nguyen, 2022), HECI (Xu et al., 2022), and LOCI (Immer et al., 2023); **(iv)** The causal velocity model CVEL (Xi et al., 2025). In Appendix D.2, we will cover additional methods (21 baselines in total based on different theories).

**Results** Table 2 shows the accuracy of QPE-k (Section 4.2) and QPE-f (Section 4.3) on 12 datasets, with more detailed and complete results available in Appendix D.2. For QPE-f, we present the results of causal flows under their best configurations, as detailed in Appendix D.1. As shown in Table 2, QPE-f performs best on these benchmarks due to its stronger expressive power and more

Table 2: Accuracy of QPE-k, QPE-f, and baselines on 12 bivariate datasets. The best is bolded.

| Method | AN | LS | SIM | SIM-c | Cha | Net | Per | Sig | Qd-V | NN-V | Tue | D4-s1 | Time (s) |
|---|---|---|---|---|---|---|---|---|---|---|---|---|---|
| ANM | 0.43 | 0.46 | 0.45 | 0.49 | 0.41 | 0.47 | 0.49 | 0.44 | 0.49 | 0.48 | 0.65 | 0.50 | 0.250 |
| PNL | 0.30 | 0.33 | 0.46 | 0.54 | 0.45 | 0.51 | 0.42 | 0.43 | 0.46 | 0.41 | 0.51 | 0.33 | 37.770 |
| RECI | 0.18 | 0.22 | 0.44 | 0.53 | 0.56 | 0.60 | 0.00 | 0.07 | 0.63 | 0.49 | 0.64 | 0.58 | 0.002 |
| CDS | 0.99 | 0.76 | 0.71 | 0.76 | 0.71 | 0.78 | 0.18 | 0.08 | 0.78 | 0.52 | 0.67 | 0.58 | 0.017 |
| CDCI | **1.00** | **1.00** | 0.84 | 0.76 | 0.67 | 0.84 | 0.48 | 0.42 | 0.74 | 0.72 | 0.68 | 0.67 | 0.001 |
| HECI | 0.98 | 0.92 | 0.49 | 0.55 | 0.57 | 0.72 | 0.01 | 0.13 | 0.59 | 0.45 | 0.61 | 0.42 | 0.026 |
| LOCI | **1.00** | **1.00** | 0.78 | 0.81 | 0.73 | 0.87 | 0.96 | 0.70 | 0.71 | 0.78 | 0.61 | 0.58 | 14.981 |
| CVEL | **1.00** | 0.98 | 0.63 | 0.72 | 0.68 | 0.62 | **1.00** | 0.84 | **0.91** | 0.87 | 0.64 | 0.67 | 1.597 |
| QPE-k | 0.99 | **1.00** | 0.83 | 0.79 | 0.60 | **0.89** | 0.77 | 0.89 | 0.42 | 0.53 | 0.54 | 0.58 | 0.009 |
| QPE-f | **1.00** | **1.00** | **0.88** | **0.88** | **0.85** | 0.86 | **1.00** | **0.90** | **0.91** | **0.90** | **0.70** | **0.79** | 7.804 |

accurate fitting, but it is relatively time-consuming due to the need to train causal flows. QPE-k performs similarly to QPE-f on various benchmarks and runs very fast, but its identification capacity is limited. Additionally, methods based on ANM or HNM do not perform as well as CVEL and QPE-f on causal flow and constrained QPE datasets. This suggests that our identifiability theories based on QPE have a broader scope of applicability beyond just common FCMs.

## 6.2 MULTIVARIATE CAUSAL ORDERING EXPERIMENTS

Table 3: Runtime efficiency of FICO and CaPS, in seconds per sub-test.

| Method | $d = 5$ | $d = 10$ | $d = 20$ | $d = 50$ | $d = 100$ |
|---|---|---|---|---|---|
| CaPS | $0.455 \pm 0.037$ | $1.074 \pm 0.056$ | $2.761 \pm 0.285$ | $10.822 \pm 1.037$ | $33.794 \pm 3.501$ |
| FICO | $0.425 \pm 0.322$ | $0.797 \pm 0.364$ | $1.727 \pm 0.523$ | $5.550 \pm 0.943$ | $13.538 \pm 1.248$ |

**Results** Given that CaPS and FICO are algorithmically equivalent, their performance is nearly identical, with minor differences attributable only to numerical computation errors. For simplicity, we therefore report only their computational efficiency in the main text.

Performance comparisons between CaPS, FICO, and other causal ordering baselines are detailed in the appendices: Appendix C.2 describes the datasets and baselines used; Appendix D.3 reports the relationship between performance and sample size; Appendix D.5 details performance on real-world datasets; and Appendix D.6 covers performance on synthetic datasets. In summary, we find that all score function based causal ordering algorithms perform very similarly across various settings, demonstrating the robustness reported by Montagna et al. (2023). This empirically suggests that the underlying assumptions or characteristics of these methods are implicitly satisfied.

Table 3 reports the average time consumed by both methods across all tests under different dimensionalities, showing that FICO significantly outperforms CaPS. For a comparison of computational efficiency between FICO and all other causal ordering methods, see Appendix D.7.

## 7 CONCLUSION

In this work, building upon research into quantile partial effects, we propose a novel parametric assumption (Assumption 3.5) that enables cause-effect identifiability (Theorem 3.6) solely from the observational distribution. This assumption simultaneously generalizes and relaxes the Functional Causal Model assumption. Consequently, we develop two algorithms, QPE-k (Section 4.2) and QPE-f (Section 4.3), for effective bivariate causal discovery, and evaluate their performance in numerous experiments (Section 6.1). For multivariate causal discovery, we investigate indirect statistical criteria related to QPE for efficient causal ordering. We propose FICO (Section 5.2), which performs causal ordering efficiently when Assumption 5.4 holds, validated on both synthetic and real datasets (Section 6.2). For causal discovery based on quantile partial effects, future work will first investigate cause-effect identifiability under more general conditions, such as relaxing the fixed basis function assumption. In multivariate causal discovery, high-dimensional quantile partial effect estimation is challenging; while this paper explores an indirect alternative, its underlying assumptions lack interpretability and practical verifiability. Future work should develop more intuitive and inherently suitable information measures for multivariate causal discovery.

ACKNOWLEDGMENTS

This work was sponsored by the Open Project of Shanghai Driverless Train Control System Engineering Technology ResearchCenter(No.SUTC-2025KT-02).

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

## A  DISCUSSION ON QPE AND IDENTIFIABILITY

### A.1  PROOFS FOR SECTION 3.2

**Proposition 3.2** (QPE from CDF). $\psi_{Y|\boldsymbol{X}} = -\nabla_{\boldsymbol{x}} F_{Y|\boldsymbol{X}} / \partial_y F_{Y|\boldsymbol{X}} = -\nabla_{\boldsymbol{x}} F_{Y|\boldsymbol{X}} / p_{Y|\boldsymbol{X}}.$

*Proof.* For any function $y = f(\boldsymbol{x}, u)$, if $f(\boldsymbol{x}, \cdot)$ is strictly monotonic for any $\boldsymbol{x}$, then its inverse is $u = \left(f^{-1}(\boldsymbol{x}, \cdot)\right)^{-1}$. Fixing $y$ and taking the total derivative w.r.t. $\boldsymbol{x}$ of the implicit function $y = f(\boldsymbol{x}, u(\boldsymbol{x}, y))$ yields

$$\frac{\mathrm{d}y}{\mathrm{d}x_i} = \partial_{x_i} f(\boldsymbol{x}, u) + \partial_u f(\boldsymbol{x}, u)\, \partial_{x_i} u(\boldsymbol{x}, y) = 0,$$

Thus, $\nabla_{\boldsymbol{x}} u = -\nabla_{\boldsymbol{x}} f / \partial_u f$. Given the inverse relationship between the conditional quantile function and the conditional CDF, $Q_{Y|\boldsymbol{X}}$ and $F_{Y|\boldsymbol{X}}$ are inverses of each other. Therefore, $\nabla_{\boldsymbol{x}} Q_{Y|\boldsymbol{X}}(\boldsymbol{x}, y) = -\nabla_{\boldsymbol{x}} F_{Y|\boldsymbol{X}} / \partial_y F_{Y|\boldsymbol{X}}$. Furthermore, from the relationship between the conditional CDF and conditional PDF, $\partial_y F_{Y|\boldsymbol{X}} = p_{Y|\boldsymbol{X}}$, which implies $\nabla_{\boldsymbol{x}} Q_{Y|\boldsymbol{X}}(\boldsymbol{x}, y) = -\nabla_{\boldsymbol{x}} F_{Y|\boldsymbol{X}} / p_{Y|\boldsymbol{X}}$. □

**Proposition 3.3** (Causal Velocity is QPE). $\psi_{Y|\boldsymbol{X}} = \nabla_{\boldsymbol{x}} f(\boldsymbol{x}, u) = -\nabla_{\boldsymbol{x}} u / \partial_y u$, *a.e.*

*Proof.* From the previous derivation, we have $\nabla_{\boldsymbol{x}} u = -\nabla_{\boldsymbol{x}} f / \partial_u f$. Fixing $\boldsymbol{x}$ and taking the total derivative w.r.t. $y$ of the implicit function $y = f(\boldsymbol{x}, u(\boldsymbol{x}, y))$ yields

$$\frac{\mathrm{d}y}{\mathrm{d}y} = \partial_u f(\boldsymbol{x}, u)\, \partial_y u(\boldsymbol{x}, y) = 1,$$

Thus, $\partial_y u = 1/\partial_u f$. Therefore, $\nabla_{\boldsymbol{x}} u / \partial_y u = -\nabla_{\boldsymbol{x}} f$, which proves the latter half of the identity.

Consider another SCM $Y = g(\boldsymbol{X}, W)$ where $W$ is uniformly distributed on $[0, 1]$, such that $g$ is strictly monotonic w.r.t. $W$ and $\boldsymbol{X} \perp\!\!\!\perp W$. Now, let any $\boldsymbol{x}$ be given. According to the probability integral transform, the conditional CDF $F_{Y|\boldsymbol{X}}$ transforms the conditional distribution $p_{Y|\boldsymbol{X}}$ into $p_W$. The inverse function $w = (g(\boldsymbol{x}, \cdot))^{-1}$ also transforms the conditional distribution $p_{Y|\boldsymbol{X}}$ into $p_W$. Since both $F_{Y|\boldsymbol{X}}$ and $w$ are strictly monotonic for continuous variables, they are both Knothe transports from $p_{Y|\boldsymbol{X}}$ to $p_W$. Due to the a.e. uniqueness of Knothe transport, $F_{Y|\boldsymbol{X}} = w$ a.e. Furthermore, as in (Nasr-Esfahany et al., 2023), under strict monotonicity and the Markov assumption, for any SCM that generates the observation distribution $P_{\boldsymbol{X},Y}$, the exogenous variables are identifiable up to an invertible transformation. Let this invertible transformation between exogenous variables $U$ and $W$ be $h$ such that $U = h(W)$. Then $\nabla_{\boldsymbol{x}} u = h' \nabla_{\boldsymbol{x}} w$ and $\partial_y u = h' \partial_y w$, which implies $\nabla_{\boldsymbol{x}} u / \partial_y u = \nabla_{\boldsymbol{x}} w / \partial_y w$. Since $F_{Y|\boldsymbol{X}} = w$ a.e., by Proposition 3.2, $\nabla_{\boldsymbol{x}} u / \partial_y u = \nabla_{\boldsymbol{x}} F_{Y|\boldsymbol{X}} / \partial_y F_{Y|\boldsymbol{X}} = -\psi_{Y|\boldsymbol{X}}$ a.e., which proves the first half of the identity. □

**Lemma 3.4.** $\nabla_{\boldsymbol{x}} \log p_{Y|\boldsymbol{X}} + \boldsymbol{\xi}\, \partial_y \log p_{Y|\boldsymbol{X}} + \partial_y \boldsymbol{\xi} = 0$ *and* $\lim_{y \to \pm\infty} \boldsymbol{\xi}\, p_{Y|\boldsymbol{X}} = 0$ *iff* $\boldsymbol{\xi} = \psi_{Y|\boldsymbol{X}}$.

*Proof.* According to Proposition 3.2, $\nabla_{\boldsymbol{x}} F_{Y|\boldsymbol{X}} + \psi_{Y|\boldsymbol{X}}\, p_{Y|\boldsymbol{X}} = 0$. Under assumptions of smoothness and strict positivity, taking the partial derivative w.r.t. $y$ gives

$$\nabla_{\boldsymbol{x}} p_{Y|\boldsymbol{X}} + \partial_y\big(\psi_{Y|\boldsymbol{X}}\, p_{Y|\boldsymbol{X}}\big) = \frac{\nabla_{\boldsymbol{x}} p_{Y|\boldsymbol{X}}}{p_{Y|\boldsymbol{X}}} + \frac{p_{Y|\boldsymbol{X}}\, \partial_y \psi_{Y|\boldsymbol{X}} + \psi_{Y|\boldsymbol{X}}\, \partial_y p_{Y|\boldsymbol{X}}}{p_{Y|\boldsymbol{X}}}$$
$$= \nabla_{\boldsymbol{x}} \log p_{Y|\boldsymbol{X}} + \psi_{Y|\boldsymbol{X}}\, \partial_y \log p_{Y|\boldsymbol{X}} + \partial_y \psi_{Y|\boldsymbol{X}} = 0, \quad (3)$$

which means the equation holds when $\boldsymbol{\xi} = \psi_{Y|\boldsymbol{X}}$. Additionally, according to Proposition 3.2 again, $\psi_{Y|\boldsymbol{X}}\, p_{Y|\boldsymbol{X}} = \nabla_{\boldsymbol{x}} F_{Y|\boldsymbol{X}}$, so

$$\lim_{y \to +\infty} \psi_{Y|\boldsymbol{X}}\, p_{Y|\boldsymbol{X}} = \lim_{y \to +\infty} \nabla_{\boldsymbol{x}} F_{Y|\boldsymbol{X}} = \nabla_{\boldsymbol{x}} \lim_{y \to +\infty} F_{Y|\boldsymbol{X}} = \lim_{y \to +\infty} 1 = 0. \quad (4)$$

The case for $y \to -\infty$ is analogous. From Equation 3 and Equation 4, the necessity of the lemma is established.

Now, for any $\boldsymbol{\xi}$ satisfying the conditions, subtract the equation it satisfies from the equation that $\psi_{Y|\boldsymbol{X}}$ satisfies to obtain

$$(\boldsymbol{\xi} - \psi_{Y|\boldsymbol{X}})\, \partial_y \log p_{Y|\boldsymbol{X}} + \partial_y(\boldsymbol{\xi} - \psi_{Y|\boldsymbol{X}}) = 0.$$

Then, multiply both sides of the equation by $p_{Y|\boldsymbol{X}}$ and expand $\partial_y \log p_{Y|\boldsymbol{X}}$, which yields

$$(\boldsymbol{\xi} - \boldsymbol{\psi}_{Y|\boldsymbol{X}}) \, \partial_y p_{Y|\boldsymbol{X}} + p_{Y|\boldsymbol{X}} \, \partial_y (\boldsymbol{\xi} - \boldsymbol{\psi}_{Y|\boldsymbol{X}}) = \partial_y \big( (\boldsymbol{\xi} - \boldsymbol{\psi}_{Y|\boldsymbol{X}}) \, p_{Y|\boldsymbol{X}} \big) = 0.$$

Integrating this w.r.t. $y$ from some $y_0$ gives

$$(\boldsymbol{\xi} - \boldsymbol{\psi}_{Y|\boldsymbol{X}}) \, p_{Y|\boldsymbol{X}} + C = 0,$$

where $C(\boldsymbol{x})$ is a function independent of $y$. Taking $y \to +\infty$ for this equation on both sides, since $\lim_{y \to +\infty} \boldsymbol{\xi} \, p_{Y|\boldsymbol{X}} = 0$ and $\lim_{y \to +\infty} \boldsymbol{\psi}_{Y|\boldsymbol{X}} \, p_{Y|\boldsymbol{X}} = 0$, it follows that $\lim_{y \to +\infty} C = C = 0$. Thus, $(\boldsymbol{\xi} - \boldsymbol{\psi}_{Y|\boldsymbol{X}}) \, p_{Y|\boldsymbol{X}} = 0$ holds for all $\boldsymbol{x}, y$. Due to the strict positivity assumption, $p_{Y|\boldsymbol{X}} > 0$, so it must be that $\boldsymbol{\xi} - \boldsymbol{\psi}_{Y|\boldsymbol{X}} = 0$, which establishes the sufficiency of the lemma. $\qquad\square$

**Theorem 3.6** (Identifiability of QPE in Finite Linear Span). *For each variable $X_i$ and any $\boldsymbol{x}$, $\psi_{Y|\boldsymbol{X},i}(\cdot \mid \boldsymbol{x}) \in span(\boldsymbol{\phi})$ implies $W_Y(s_{X_i,Y}, \boldsymbol{\eta}_{Y|\boldsymbol{X}}) = 0$ for any $y$. If: (i) The components in $\boldsymbol{\eta}_{Y|\boldsymbol{X}}$ are linearly independent; (ii) There exists $y$ such that $W_Y(\boldsymbol{\eta}_{Y|\boldsymbol{X}}) \neq 0$; (iii) For each basis function $\phi_j$, $\lim_{y \to \pm\infty} \phi_j \, p_{Y|\boldsymbol{X}} = 0$; then the converse is also true.*

*Proof.* Let an arbitrary $\boldsymbol{x}$ be given in this proof. Substituting $\psi_{Y|\boldsymbol{X},i}(y \mid \boldsymbol{x}) = \sum_{j=1}^{k} c_{i,j}(\boldsymbol{x}) \, \phi_j(y)$ into the second-order cross PDE (Equation 2) and rearranging terms yields

$$\partial_{x_i} \partial_y \log p_{\boldsymbol{X},Y} + \sum_{j=1}^{k} c_{i,j} \, \partial_y \big( \phi_j \, \partial_y \log p_{\boldsymbol{X},Y} + \phi_j' \big) = s_{X_i,Y} + \sum_{j=1}^{k} c_{i,j} \, \eta_{Y|\boldsymbol{X},j} = 0, \qquad (5)$$

where $\eta_{Y|\boldsymbol{X},j}$ denotes the $j$-th component of $\boldsymbol{\eta}_{Y|\boldsymbol{X}}$. This implies that $s_{X_i,Y}$ and $\boldsymbol{\eta}_{Y|\boldsymbol{X}}$ are linearly dependent. Taking the $m$-th partial derivative w.r.t. $y$ for $m = 0, \dots, k$ gives

$$\partial_y^m s_{X_i,Y} + \sum_{j=1}^{k} c_{i,j} \, \partial_y^m \eta_{Y|\boldsymbol{X},j} = 0,$$

for all $m = 0, \dots, k$. Note that these $k+1$ equations (including the zero-th order) share the same coefficients. Therefore,

$$\begin{bmatrix} s_{X_i,Y} & \eta_{Y|\boldsymbol{X},1} & \cdots & \eta_{Y|\boldsymbol{X},k} \\ \partial_y s_{X_i,Y} & \partial_y \eta_{Y|\boldsymbol{X},1} & \cdots & \partial_y \eta_{Y|\boldsymbol{X},k} \\ \vdots & \vdots & \ddots & \vdots \\ \partial_y^k s_{X_i,Y} & \partial_y^k \eta_{Y|\boldsymbol{X},1} & \cdots & \partial_y^k \eta_{Y|\boldsymbol{X},k} \end{bmatrix} \begin{bmatrix} 1 \\ c_{i1} \\ \vdots \\ c_{ik} \end{bmatrix} = \boldsymbol{W} \boldsymbol{c}^{\mathsf{T}} = 0,$$

for any $y$. Since $\boldsymbol{c}$ is not identically zero, $\det \boldsymbol{W} = 0$, which means $W_Y(s_{X_i,Y}, \boldsymbol{\eta}_{Y|\boldsymbol{X}}) = 0$.

Assume $W_Y(s_{X_i,Y}, \boldsymbol{\eta}_{Y|\boldsymbol{X}}) = 0$ for any $y$. By the Peano-Bôcher Theorem (Bocher, 1901), the two additional conditions **(i)** and **(ii)** guarantee that $s_{X_i,Y}$ and $\boldsymbol{\eta}_{Y|\boldsymbol{X}}$ are linearly dependent. Thus, there exist $1, c_{i,1}, \dots, c_{ik}$ such that Equation 2 holds. We only consider the zero-th order equation, i.e., Equation 5 holds. Integrating it w.r.t. $y$ from some $y_0$ yields

$$\partial_{x_i} \log p_{\boldsymbol{X},Y} + \sum_{j=1}^{k} c_{i,j} \big( \phi_j \, \partial_y \log p_{\boldsymbol{X},Y} + \phi_j' \big) + C = 0, \qquad (6)$$

where $C(\boldsymbol{x})$ is a function independent of $y$. Now, taking the expectation of both sides w.r.t. $p_{Y|\boldsymbol{X}}$, notice that

$$\mathbb{E}\left[ \partial_{x_i} \log p_{\boldsymbol{X},Y} \mid \boldsymbol{X} = \boldsymbol{x} \right] = \frac{1}{p_{\boldsymbol{X}}} \int \partial_{x_i} p_{\boldsymbol{X},Y} \, \mathrm{d}y = \partial_{x_i} \log p_{\boldsymbol{X}}.$$

Furthermore, $\phi_j \, \partial_y \log p_{\boldsymbol{X},Y} + \phi_j'$ is the Stein operator acting on $\phi_j$ w.r.t. $p_{Y|\boldsymbol{X}}$ (since $\partial_y \log p_{Y|\boldsymbol{X}} = \partial_y \log p_{\boldsymbol{X},Y}$). Thus, by Stein's identity (under the vanishing condition **(iii)**),

$$\mathbb{E}\left[ \phi_j \, \partial_y \log p_{\boldsymbol{X},Y} + \phi_j' \mid \boldsymbol{X} = \boldsymbol{x} \right] = 0,$$

for each $j$. Also, $\mathbb{E}\left[ C(\boldsymbol{x}) \mid \boldsymbol{X} = \boldsymbol{x} \right] = C(\boldsymbol{x})$. For Equation 2 to hold, $C = -\partial_{x_i} \log p_{\boldsymbol{X}}$. Rearranging Equation 6 further, we get

$$\partial_{x_i} \log p_{Y|\boldsymbol{X}} + \left( \sum_{j=1}^{k} c_{i,j} \, \phi_j \right) \partial_y \log p_{Y|\boldsymbol{X}} + \partial_y \left( \sum_{j=1}^{k} c_{i,j} \, \phi_j \right) = 0.$$

Given the boundary condition $\lim_{y \to +\infty} \phi_j \, p_{Y|X} = 0$, let each component of $\boldsymbol{\xi}$ be $\xi_i = \sum_{j=1}^{k} c_{i,j} \, \phi_j$. Then

$$\lim_{y \to +\infty} \xi_i \, p_{Y|X} = \sum_{j=1}^{k} c_{i,j} \lim_{y \to +\infty} \phi_j \, p_{Y|X} = 0.$$

Therefore, by the equivalence stated in Lemma 3.4, $\xi_i = \psi_{Y|X,i}$, i.e., $\psi_{Y|X,i}(\cdot \,|\, \boldsymbol{x}) \in \operatorname{span}(\boldsymbol{\phi})$. $\quad\square$

## A.2 PROOFS FOR SECTION 4.1

**Corollary 4.2.** *Assume that there are $k$ basis functions in $\boldsymbol{\phi}$ and certain regularity conditions hold. Then: (i)* $\dim(\boldsymbol{\Theta}_{X \to Y; \boldsymbol{\phi}}) = \infty$*; (ii)* $\dim(\boldsymbol{\Theta}_{X \leftarrow Y; \boldsymbol{\phi}}) \leq k + 2$.

*Proof.* Here, we can consider the log-likelihood $\log p_X$, given the assumption of absolute positivity. First, by Markovianity and strict monotonicity, there exists an equation by change-of-variable formula:

$$\log p_{Y|X} = \log p_U - \log \partial_u f = \log p_U + \log \partial_y u,$$

where $U$ can be expressed by $X, Y$, and $f$. Thus, $\log p_{Y|X}$ only depends on $f$ and $p_U$, which are assumed to be given. So $\log p_{Y|X}$ is a known function. Since the joint log-density is $\log p_{X,Y} = \log p_X + \log p_{Y|X}$, for any given $y_0$, any PDE on $\log p_{X,Y}$ can be simplified to an ODE solely on $\log p_X$. Furthermore, any partial or mixed derivative w.r.t. $y$ will only involve the given $\log p_{Y|X}$.

Next, when the forward version of Assumption 3.5 holds, the Wronskian determinant described in Theorem 3.6 is an identity involving only partial or mixed derivatives w.r.t. $y$. This is completely independent of $\log p_X$. In other words, the forward version of Assumption 3.5 does not impose new constraints, which implies $\boldsymbol{\Theta}_{X \to Y; \boldsymbol{\phi}} = \boldsymbol{\Theta}$, and thus $\dim(\boldsymbol{\Theta}_{X \to Y; \boldsymbol{\phi}}) = \infty$.

Conversely, when the backward version of Assumption 3.5 holds, the Wronskian determinant in Theorem 3.6 requires an ODE for $\log p_X$ to be satisfied. This ODE can be rewritten in the form

$$(\log p_X)^{(k+2)} = G(x, \log p_X, (\log p_X)', \dots, (\log p_X)^{(k+1)}),$$

whose highest order is $k + 2$. This high-order nonlinear ODE can be reduced to a $k + 2$-dimensional first-order ODE system. According to the Picard-Lindelöf theorem, under certain regularity conditions (e.g., global Lipschitz continuity), the solution to this ODE exists and is unique, determined by $k + 2$ initial conditions $(\log p_X(x_0), (\log p_X)'(x_0), \dots, (\log p_X)^{(k+1)}(x_0))$. Therefore, $\log p_X$ can be described by these $k + 2$ parameters. If degenerate cases exist, fewer parameters may be required. Hence, $\dim(\boldsymbol{\Theta}_{X \leftarrow Y; \boldsymbol{\phi}}) \leq k + 2$. $\quad\square$

**Discussion on Corollary 4.2** As stated in the main text, this corollary generalizes the conclusion by (Hoyer et al., 2008) that unidentifiable ANMs lie in a three-dimensional affine space (since ANMs correspond to $k = 1$ and the Wronskian simplifies to a linear ODE). It also fills the gap left by previous identifiability works (Immer et al., 2023; Strobl & Lasko, 2023) regarding quantifying unidentifiable HNMs (HNMs correspond to $k = 2$). Similar to our work, Xi et al. (2025) also derived identifiability ODEs for ANMs and HNMs using causal velocity, but they did not further quantify the solutions for HNMs. In contrast to these works, Corollary 4.2 provides quantified identifiability for HNMs and extends this conclusion to any finite-linearly spanned QPE ($k < \infty$).

## A.3 QPE IS IRRELEVANT TO MONOTONIC MECHANISM AND MARKOV PROPERTY

(Xi et al., 2025) demonstrated that causal velocity, specifically QPE as in Proposition 3.3, is independent of the precise form of noise and latent mechanisms. This relaxes the strength of parametric assumptions on FCMs. Given that QPE is an observational quantity, inferable solely from observational distributions, we have relaxed the counterfactual assumptions typically required for causal velocity, including strict monotonic mechanisms and the Markov assumption, which are crucial for counterfactual identification (Nasr-Esfahany et al., 2023; Chen & Du, 2025). Therefore, as claimed in the main text, QPE fully relaxes assumptions regarding the latent causal mechanisms, shifting focus entirely to the observational distribution's shape. This implies that "observational causal discovery," even requiring identifiability of causal direction, can operate entirely within the observational layer of the causal hierarchy.

Next, we provide two simple examples demonstrating that even if the underlying causal mechanisms do not satisfy strict monotonicity or the Markov property, their observational distributions' QPEs are identical and still satisfy Assumption 3.5:

**Example A.1** (Strictly Monotonic Mechanism but Semi-Markovian). Let $X = Z + W$ and $Y = \exp(X^2 + Z + U)$, where the confounding variable $Z$ and the exogenous variables $U, W$ are independent standard normal. Then the QPE $\psi_{Y|X}(y\,|\,x) = -(2x + 0.5)y$.

*Proof.* Utilizing the property of linear Gaussians, consider $Z = X - W$. Then $(Z \mid X = x) \sim \mathcal{N}(0.5x, 0.5)$, and simultaneously $(Z + U \mid X = x) \sim \mathcal{N}(0.5x, 1.5)$. Let $T \sim \mathcal{N}(0, 1.5)$, so $Z + U \mid X = x$ is equivalent to $T + 0.5x$. Thus, the conditional event $Y \leq y \mid X = x$ holds if and only if $x^2 + T + 0.5x \leq \log(y)$. Therefore, $F_{Y|X}(y|x) = P(Y \leq y \mid X = x) = P(T \leq \log(y) - x^2 - 0.5x) = F_T(\log(y) - x^2 - 0.5x) = h(x, y)$. We can calculate $\partial_x F_T = h(x, y)(-2x - 0.5)$ and $\partial_y F_T = h(x, y)(1/y)$, which are $\partial_x F_{Y|X}(y|x)$ and $\partial_y F_{Y|X}(y|x)$ respectively. By Proposition 3.2, we obtain $\psi_{Y|X}(y\,|\,x) = -(2x + 0.5)y$. $\qquad\square$

**Example A.2** (Non-Monotonic Mechanism and Semi-Markovian). Let $X = Z + W$ and $Y = \exp(X + Z + U^2)$, where the confounding variable $Z$ and the exogenous variables $U, W$ are independent standard normal. Then the QPE $\psi_{Y|X}(y\,|\,x) = -1.5y$.

*Proof.* Utilizing the property of linear Gaussians, consider $Z = X - W$. Then $(Z \mid X = x) \sim \mathcal{N}(0.5x, 0.5)$. Since $U^2 \sim \chi_1^2$ and is independent of $Z$ and $X$, $(Z + U^2 \mid X = x) \sim \mathcal{N}(0.5x, 0.5) + \chi_1^2$. Let $T \sim \mathcal{N}(0, 0.5) + \chi_1^2$, then $Z + U^2 \mid X = x$ is equivalent to $T + 0.5x$. Thus, the conditional event $Y \leq y \mid X = x$ holds if and only if $x + T + 0.5x \leq \log(y)$. Therefore, $F_{Y|X}(y|x) = P(Y \leq y \mid X = x) = P(T \leq \log(y) - 1.5x) = F_T(\log(y) - 1.5x) = h(x, y)$. We can calculate $\partial_x F_T = -1.5h(x, y)$ and $\partial_y F_T = h(x, y)(1/y)$, which are $\partial_x F_{Y|X}(y|x)$ and $\partial_y F_{Y|X}(y|x)$ respectively. By Proposition 3.2, we obtain $\psi_{Y|X}(y\,|\,x) = -1.5y$. $\qquad\square$

In Example A.1, $Z + U \mid X = x$ correlates with $x$, implying that the non-endogenous term $\exp(Z + U)$ and the endogenous term $\exp(X)$ are not independent, thus violating the Markov property. However, since it can be equivalently written as some random variable $T + 0.5x$, where $T$ is conveniently eliminated when computing QPE, it "coincidentally" still satisfies Assumption 3.5. This means its observational distribution can even be equivalent to an HNM (the basis is $y$). In Example A.2, the principle for violating the Markov property is similar to Example A.1, with the difference being that the non-endogenous term $\exp(Z + U^2)$ is non-monotonic w.r.t the exogenous variable. Yet, it can still be equivalently written as some random variable $T + 0.5x$. The existence of these two examples reveals a series of special cases where, even if the underlying SCM does not satisfy strict monotonicity or the Markov property, its observational distribution still exhibits particular characteristics in QPE. In such cases, assumptions on FCMs fail, but assumptions on QPE remain valid.

It is crucial to emphasize that despite QPE being independent of the underlying causal mechanisms, it still represents a parametric assumption. This parametric assumption is empirically verifiable when the analytical form of the observational distribution is fully known. For instance, we can compute the Wronskian determinant given in Theorem 3.6 to empirically determine if this assumption holds. However, in reality, one typically only has samples from the observational distribution. Therefore, whether the QPE parametric assumption holds still requires careful consideration in real-world problems, similar to FCMs, to mitigate potential risks arising from assumption violations.

## B DISCUSSION ON FICO

### B.1 PROOFS FOR SECTION 5.1

**Theorem 5.2.** *For each covariate $X_i$, if Assumption 5.1 holds then*

$$\mathbb{E}\left[(\psi_{Y|\boldsymbol{X},i})^2 (s_Y)^2\right] = \mathbb{E}\left[(s_{X_i})^2\right] - \mathbb{E}\left[(r_{X_i})^2\right] + \mathbb{E}\left[(\partial_y \psi_{Y|\boldsymbol{X},i})^2 + 2\,\psi_{Y|\boldsymbol{X},i}\,\partial_y^2 \psi_{Y|\boldsymbol{X},i}\right].$$

*Proof.* Rewrite the equation described in Lemma 3.4 as:

$$s_{X_i} = r_{X_i} - \psi\,s_Y - \partial_y \psi,$$

where $\psi_{Y|\boldsymbol{X},i}$ is abbreviated as $\psi$. Squaring both sides and taking the expectation, the following equation still holds:

$$\mathbb{E}\left[(s_{X_i})^2\right] = \mathbb{E}\left[(r_{X_i})^2\right] + \mathbb{E}\left[(\psi\,s_Y + \partial_y \psi)^2\right] - 2\,\mathbb{E}\left[r_{X_i}(\psi\,s_Y + \partial_y \psi)\right], \tag{7}$$

where $\mathbb{E}\left[r_{X_i}(\psi\,s_Y + \partial_y \psi)\right] = \mathbb{E}\left[r_{X_i}\,\mathbb{E}\left[\psi\,s_Y + \partial_y \psi \mid \boldsymbol{X} = \boldsymbol{x}\right]\right]$. And $\mathbb{E}\left[\psi\,s_Y + \partial_y \psi \mid \boldsymbol{X} = \boldsymbol{x}\right] = 0$ due to Stein's identity, assuming $\lim_{y\to\infty} \psi\,p_{\boldsymbol{X},Y} = 0$. Thus, the entire equation simplifies to:

$$\mathbb{E}\left[(s_{X_i})^2\right] = \mathbb{E}\left[(r_{X_i})^2\right] + \mathbb{E}\left[(\psi\,s_Y + \partial_y \psi)^2\right].$$

Now consider $\mathbb{E}\left[(\psi\,s_Y + \partial_y \psi)^2\right] = \mathbb{E}\left[\psi^2 (s_Y)^2\right] + \mathbb{E}\left[(\partial_y \psi)^2\right] + 2\,\mathbb{E}\left[s_Y\,\psi\,\partial_y \psi\right]$. Also, according to Stein's identity, if $\lim_{y\to\infty} \psi\,\partial_y \psi\,p_{\boldsymbol{X},Y} = 0$, then

$$\mathbb{E}\left[s_Y\,\psi\,\partial_y \psi + \partial_y(\psi\,\partial_y \psi)\right] = 0.$$

Therefore, $\mathbb{E}\left[s_Y\,\psi\,\partial_y \psi\right] = -\mathbb{E}\left[(\partial_y \psi)^2 + \psi\,\partial_y^2 \psi\right]$. Substituting these simplified terms back into Equation 7 yields the theorem. $\qquad\square$

**Corollary 5.3.** *For each covariate $X_i$, if Assumption 5.1 holds then*

$$\mathbb{E}\left[(\partial_y \psi_{Y|\boldsymbol{X},i})^2 + 2\,\psi_{Y|\boldsymbol{X},i}\,\partial_y^2 \psi_{Y|\boldsymbol{X},i}\right] < \mathbb{E}\left[((\psi_{Y|\boldsymbol{X},i})^2 - 1)(s_Y)^2\right] + \mathbb{E}\left[(r_{X_i})^2\right]$$

*if and only if $\mathbb{E}\left[(s_{X_i})^2\right] > \mathbb{E}\left[(s_Y)^2\right]$.*

*Proof.* According to Theorem 5.2, $\mathbb{E}\left[(s_{X_i})^2\right] - \mathbb{E}\left[(s_Y)^2\right]$ equals

$$\mathbb{E}\left[(r_{X_i})^2\right] + \mathbb{E}\left[((\psi_{Y|\boldsymbol{X},i})^2 - 1)(s_Y)^2\right] - \mathbb{E}\left[(\partial_y \psi_{Y|\boldsymbol{X},i})^2 + 2\,\psi_{Y|\boldsymbol{X},i}\,\partial_y^2 \psi_{Y|\boldsymbol{X},i}\right].$$

Therefore, this inequality holds iff $\mathbb{E}\left[(s_{X_i})^2\right] - \mathbb{E}\left[(s_Y)^2\right] > 0$, which proves the corollary. $\qquad\square$

### B.2 FICO UNDER HETEROSCEDASTIC GAUSSIAN ASSUMPTION

First, we provide the formal statement and proof of the corollary to Corollary 5.3 regarding the HNM assumption, as discussed in the main text:

**Corollary B.1.** *For each covariate $X_i$, given Assumption 5.1 holds. If $\partial_y^2 \psi_{Y|\boldsymbol{X},i} = 0$ and*

$$\mathbb{E}\left[(\psi_{Y|\boldsymbol{X},i})^2\right] > 1 - \frac{\mathbb{E}\left[(r_Y)^2\right]}{\mathbb{E}\left[(s_Y)^2\right]} + \frac{\mathbb{E}\left[(\partial_y \psi_{Y|\boldsymbol{X},i})^2\right] + \sqrt{Var\left[(\psi_{Y|\boldsymbol{X},i})^2\right] Var\left[(s_Y)^2\right)\right]}}{\mathbb{E}\left[(s_Y)^2\right]},$$

*then $\mathbb{E}\left[(s_{X_i})^2\right] > \mathbb{E}\left[(s_Y)^2\right]$.*

*Proof.* When $\partial_y^2 \psi = 0$, $\mathbb{E}\left[(s_{X_i})^2\right] - \mathbb{E}\left[(s_Y)^2\right]$ equals

$$\mathbb{E}\left[(r_{X_i})^2\right] + \mathbb{E}\left[(\psi^2 - 1)(s_Y)^2\right] - \mathbb{E}\left[(\partial_y \psi)^2\right],$$

where

$$\mathbb{E}\left[\psi^2 (s_Y)^2\right] = \mathbb{E}\left[\psi^2\right] \mathbb{E}\left[(s_Y)^2\right] + \text{Cov}\left(\psi^2, (s_Y)^2\right).$$

And according to the Cauchy-Schwarz inequality,

$$\text{Cov}\left(\psi^2, (s_Y)^2\right) \geq -\sqrt{\text{Var}\left[\psi^2\right] \text{Var}\left[(s_Y)^2\right]}.$$

Assuming $\mathbb{E}\left[(s_Y)^2\right] > 0$, then follow the condition in this corollary,

$$\mathbb{E}\left[(r_{X_i})^2\right] + \mathbb{E}\left[\psi^2\right]\mathbb{E}\left[(s_Y)^2\right] - \mathbb{E}\left[(\partial_y\psi)^2\right] - \sqrt{\operatorname{Var}\left[\psi^2\right]\operatorname{Var}\left[(s_Y)^2\right)\right]} > \mathbb{E}[(s_Y)^2].$$

Since the inequality for $\operatorname{Cov}\left(\psi^2, (s_Y)^2\right)$ holds, it implies

$$\mathbb{E}\left[(r_{X_i})^2\right] + \mathbb{E}\left[\psi^2\right]\mathbb{E}\left[(s_Y)^2\right] - \mathbb{E}\left[(\partial_y\psi)^2\right] + \operatorname{Cov}\left(\psi^2, (s_Y)^2\right) > \mathbb{E}[(s_Y)^2].$$

Rearranging this gives

$$\mathbb{E}\left[(r_{X_i})^2\right] + \mathbb{E}\left[(\psi^2 - 1)(s_Y)^2\right] - \mathbb{E}\left[(\partial_y\psi)^2\right] = \mathbb{E}\left[(s_{X_i})^2\right] - \mathbb{E}\left[(s_Y)^2\right] > 0,$$

which completes the proof. $\qquad\square$

Corollary B.1 qualitatively requires the squared coefficient of variation (CV) of the QPE to be as small as possible. Next, we show how to derive the simplified explanation presented in the main text under the linear heteroscedastic Gaussian assumption, and how to extend this to the nonlinear case to obtain a qualitative explanation. Without loss of generality, for simplicity, we only consider the bivariate case below. The multivariate case only requires replacing the univariate functions or their derivatives in this section with multivariate functions or their partial derivatives.

**Assumption B.2.** The conditional distribution $p_{Y|X}$ is Gaussian for any $x$, i.e., $Y \mid X = x \sim \mathcal{N}(\mu(x), (\sigma(x))^2)$.

Using the expression for the Gaussian distribution and Proposition 3.2, it is straightforward to derive the QPE $\psi_{Y|X} = -\mu' - (y - \mu)(\sigma'/\sigma)$. For any $x$, $Z = (Y - \mu)/\sigma \sim \mathcal{N}(0, 1)$, so $\psi_{Y|X} = -\mu' - Z\sigma'$. Thus, it can be derived that

$$\mathbb{E}\left[(\psi_{Y|X})^2 \mid X = x\right] = \mathbb{E}\left[(\mu' + Z\sigma')^2\right] = (\mu')^2 + (\sigma')^2,$$
$$\mathbb{E}\left[(\psi_{Y|X})^4 \mid X = x\right] = \mathbb{E}\left[(\mu' + Z\sigma')^4\right] = (\mu')^4 + 6(\mu')^2(\sigma')^2 + 3(\sigma')^4,$$

where $x$ is given, so $\mu'$ and $\sigma'$ are treated as constants. For a standard normal variable $Z$, its moments are $\mathbb{E}[Z] = 0, \mathbb{E}[Z^2] = 1, \mathbb{E}[Z^3] = 0, \mathbb{E}[Z^4] = 3$. This further leads to the mean and variance of the squared QPE as:

$$\mathbb{E}\left[(\psi_{Y|X})^2\right] = \mathbb{E}\left[\mathbb{E}\left[(\psi_{Y|X})^2 \mid X = x\right]\right] = \mathbb{E}\left[(\mu')^2 + (\sigma')^2\right],$$
$$\operatorname{Var}\left[(\psi_{Y|X})^2\right] = \mathbb{E}\left[\operatorname{Var}\left[(\psi_{Y|X})^2 \mid X = x\right]\right] + \operatorname{Var}\left[\mathbb{E}\left[(\psi_{Y|X})^2 \mid X = x\right]\right]$$
$$= \mathbb{E}\left[\mathbb{E}\left[(\psi_{Y|X})^4 \mid X = x\right] - \mathbb{E}^2\left[(\psi_{Y|X})^2 \mid X = x\right]\right] + \operatorname{Var}\left[(\mu')^2 + (\sigma')^2\right]$$
$$= \mathbb{E}\left[((\mu')^4 + 6(\mu')^2(\sigma')^2 + 3(\sigma')^4) - ((\mu')^2 + (\sigma')^2)^2\right] + \operatorname{Var}\left[(\mu')^2 + (\sigma')^2\right]$$
$$= 2\,\mathbb{E}\left[2(\mu')^2(\sigma')^2 + (\sigma')^4\right] + \operatorname{Var}\left[(\mu')^2 + (\sigma')^2\right],$$

by the law of total expectation and law of total variance. Let $\kappa = \sigma'/\mu'$. Then the mean and variance can be written as:

$$\mathbb{E}\left[(\psi_{Y|X})^2\right] = \mathbb{E}\left[(\mu')^2(1 + \kappa^2)\right],$$
$$\operatorname{Var}\left[(\psi_{Y|X})^2\right] = 2\,\mathbb{E}\left[(\mu')^4\,\kappa^2\,(2 + \kappa^2)\right] + \operatorname{Var}\left[(\mu')^2(1 + \kappa^2)\right].$$

When $\mu$ and $\sigma$ are linear functions, $\mu'$ and $\sigma'$ are constants w.r.t. $X$, and thus $\kappa$ is also a constant. In this case, the CV simplifies to

$$\frac{\sqrt{\operatorname{Var}\left[(\psi_{Y|X})^2\right]}}{\mathbb{E}\left[(\psi_{Y|X})^2\right]} = \frac{\sqrt{4 + 2\kappa^2}}{1 + \kappa^2}|\kappa|. \tag{8}$$

Note that this function strictly increases as $|\kappa| \to \infty$ and its limit is $\sqrt{2}$. Therefore, to minimize the CV under the linear heteroscedastic Gaussian assumption, $|\kappa|$, i.e., $|\sigma'/\mu'|$, should be as small as possible. This means either $\sigma'$ is as small as possible (e.g., degenerating to ANM when $\sigma' = 0$) or $\mu'$ is sufficiently large relative to $\sigma'$ such that $\mu$ dominates the distribution shape.

When $\mu$ and $\sigma$ are general, assuming $l \leq |\kappa| \leq u$. We can still extract $|\kappa|$ from the moments of QPE, yielding

$$\mathbb{E}\left[(\psi_{Y|X})^2\right] \geq (1 + l^2)\, \mathbb{E}\left[(\mu')^2\right],$$
$$\mathrm{Var}\left[(\psi_{Y|X})^2\right] \leq 2\,\mathbb{E}\left[(\mu')^4\, \kappa^2\,(2 + \kappa^2)\right] + \mathbb{E}\left[((\mu')^2(1 + \kappa^2))^2\right]$$
$$\leq (1 + 6u^2 + 3u^4)\,\mathbb{E}\left[(\mu')^4\right]$$

Substituting this into the CV expression, we obtain an upper bound for the CV:

$$\frac{\sqrt{\mathrm{Var}\left[(\psi_{Y|X})^2\right]}}{\mathbb{E}\left[(\psi_{Y|X})^2\right]} \leq \left(\frac{\sqrt{1 + 6u^2 + 3u^4}}{1 + l^2}\right)\frac{\sqrt{\mathbb{E}\left[(\mu')^4\right]}}{\mathbb{E}\left[(\mu')^2\right]}. \tag{9}$$

To minimize the CV, the upper bound must be minimized. This requires two conditions: **(i)** The upper bound of $|\kappa|$ should be as small as possible, and the lower bound as large as possible. This means $|\kappa|$ should not vary significantly (it is constant under the linear assumption). Additionally, because the growth rate of the numerator's upper bound is greater than that of the denominator's lower bound, $|\kappa|$ should generally be as small as possible. **(ii)** $\sqrt{\mathbb{E}\left[(\mu')^4\right]}/\mathbb{E}\left[(\mu')^2\right]$ should be sufficiently small. According to the Cauchy-Schwarz inequality, its lower bound is $1$ (with equality when $\mu'$ is constant, i.e., under the linear assumption).

## C  EXPERIMENT DETAILS AND RELATED WORKS

### C.1  BIVARIATE CAUSAL DISCOVERY

**Datasets**  Section 6.1 enumerates the bivariate causal discovery datasets used in our experiments, along with their sources. Most of these datasets follow conventions from previous work (Immer et al., 2023; Xi et al., 2025) and are standardized. For the Tübingen cause-effect pairs challenge, we manually removed discrete pairs with IDs "47, 52, 53, 54, 55, 70, 71, 105, 107". These pairs visibly and severely violate the continuity assumption of distributions, which most bivariate causal discovery algorithms rely on. This selection may differ from previous literature, potentially leading to lower baseline results reported in this paper.

Additionally, we created a set of constrained QPE synthetic datasets, including Qd-V, Sig-V, Rbf-V, and NN-V. These datasets are generated from the random optimization of the following objective:

$$\arg\min_{\phi,\theta} \left\| \psi_{Y|\boldsymbol{X},\phi} + \frac{\partial_{\boldsymbol{x}} u_\theta}{\partial_y u_\theta} \right\| + \lambda \mathrm{Var}\left[ \psi_{Y|\boldsymbol{X},\phi} \right],$$

where $u_\theta$ is a randomly initialized causal flow (see Equation 1), and we choose a random Gaussian mixture (3 components, with variance $0.25 \leq \sigma \leq 2.0$ for each component) as the exogenous distribution $p_U$ for the causal flow. $\psi_{Y|\boldsymbol{X},\phi}$ is a constrained QPE satisfying Assumption 3.5 (i.e., given basis functions), randomly initialized with parameters $\phi$ (which control the coefficient functions). Qd-V, Sig-V, Rbf-V, and NN-V correspond to quadratic basis $(1, y, y^2)$, sigmoid basis $(1, \mathrm{sigmoid}(y), \mathrm{sigmoid}(2y))$, RBF basis $(1, \exp(-y^2), \exp(-(y + 0.5)^2), \exp(-(y - 0.5)^2))$, and neural network basis (with $\tanh$ activation function), respectively. $\lambda$ is a regularization parameter, set to $\lambda = 0.1$, to prevent the random optimization from making the QPE too extreme, which could lead to excessively irregular observational distributions.

**Baselines**  In the main text, we primarily use 8 baseline methods as examples. In Appendix D.2, we provide a comprehensive evaluation using 21 open-source baselines based on various theories to fully and transparently demonstrate the superiority of our method. These baselines include:

- **Linear Non-Gaussian Acyclic Model**: ICA-LiNGAM (Shimizu et al., 2006), VAR-LiNGAM (Hyvärinen et al., 2010), Direct-LiNGAM (Shimizu et al., 2011).
- **Additive Noise Model**: ANM (Hoyer et al., 2008), CAM (Bühlmann et al., 2014), RESIT (Peters et al., 2014), RECI (Bloebaum et al., 2018), CGNN (Goudet et al., 2018), CDS (Fonollosa, 2019).
- **Post Non-Linear Model**: PNL (Zhang & Hyvärinen, 2009).
- **Heteroscedastic Noise Model**: QCCD (Tagasovska et al., 2020), CAREFL (Khemakhem et al., 2021), HECI (Xu et al., 2022), GRCI (Strobl & Lasko, 2023), CDCI (Duong & Nguyen, 2022), LOCI (Immer et al., 2023).
- **Minimum Description Length**: IGCI (Daniušis et al., 2010), SLOPE (Marx & Vreeken, 2019a), SLOPPY (Marx & Vreeken, 2019b).
- **Optimal Transport**: DIVOT (Tu et al., 2022).
- **Causal Velocity Model**: CVEL (Xi et al., 2025).

Among these methods, SLOPPY includes variants using different information criteria (AIC and BIC). CAREFL and LOCI include variants using likelihood and HSIC (Hilbert-Schmidt Independence Criterion) (Immer et al., 2023; Sun & Schulte, 2023). CDCI and CVEL involve multiple configurations or combinations. For all these methods, we tuned their hyperparameters to select the best configuration for a fair comparison with our method. However, we only tested the ANM variant of DIVOT due to its deprecated dependencies for PNL implementation.

While several methods share our goal of generalizing the functional forms of LiNGAM, ANM, or HNM, they achieve identifiability via assumptions that differ from our setting, such as requirements on the data generation process (Guo et al., 2023), contrastive learning frameworks (Reizinger et al., 2023), or access to multi-domain data (Jalaldoust et al., 2025). In contrast, our method, QPE, identifies the causal direction using only the observational distribution. Accordingly, our main comparisons are with methods designed for this purely observational context.

**Runtime**   Regarding hardware, PNL, CGNN, CAREFL, LOCI, CVEL, and QPE-f utilize GPU acceleration for neural networks, while others run on CPU. As for environments, implementations of SLOPE, SLOPPY, QCCD, and RECI are based on R, while others are based on Python. Each experiment is conducted using the same hardware, system, and default runtime configurations. The average time taken for one cause-effect pair under these conditions has been reported in Table 2.

**Complexity analysis**   For QPE-k, we can analyze its complexity. Let $N$ be the number of samples, $M$ be the number of test locations, and $T$ be the number of test samples used in the OLS test. The most computationally intensive part is the estimation of the response matrix in OLS, which has a complexity of $\mathcal{O}(NMT)$. In our experiments, we set $M = T = 20$, and test locations are uniformly distributed within $[-2.5, 2.5]$ since the datasets are standardized.

For QPE-f, which is based on neural networks and random optimization, its training process is simpler compared to CVEL involving higher order terms. QPE-f only needs to be trained according to Equation 1. Although calculating $\log |\partial_{x_i} u_\theta|$ also involves gradients, it can be expressed in closed form using pre-defined discrete transformations of the causal flow, as detailed in Appendix D.1.

## C.2   Multivariate Causal Ordering

**Datasets**   The configurations for the synthetic datasets are detailed as follows:

- ANM-GP and HNM-GP are generated by the following processes, respectively:

$$X_i = \text{GP}(\boldsymbol{P}_i; \boldsymbol{\theta}_i) + \text{GMM}(\boldsymbol{\mu}_i, \boldsymbol{\sigma}_i), \quad X_i = \text{GP}(\boldsymbol{P}_i; \boldsymbol{\theta}_{i,1}) + \text{GP}(\boldsymbol{P}_i; \boldsymbol{\theta}_{i,2}) \cdot \text{GMM}(\boldsymbol{\mu}_i, \boldsymbol{\sigma}_i),$$

  where $\boldsymbol{P}_i$ denotes the parent variables in the DAG, GP is the random fourier features Gaussian process, and GMM is a Gaussian mixture model. All parameters, including $\boldsymbol{\theta}_i, \boldsymbol{\theta}_{i,1}, \boldsymbol{\theta}_{i,2}, \boldsymbol{\mu}_i, \boldsymbol{\sigma}_i$, are randomly selected according to a prior.
- The gCastle (Zhang et al., 2021) synthetic process can be represented as $X_i = \text{NN}(\boldsymbol{P}_i; \boldsymbol{\theta}_i) + \mathcal{N}(0, 1)$, where NN is a neural network mechanism, $\mathcal{N}(0, 1)$ is the standard normal distribution, and the neural network parameters are randomly chosen.
- The LiNGAM (Shimizu et al., 2006) synthetic process can be represented as $X_i = \boldsymbol{a}_i^\mathsf{T} \boldsymbol{P}_i + b_i + \text{Gumbel}(0, 1)$, where $\text{Gumbel}(0, 1)$ is the standard Gumbel distribution, and the coefficients and the bias of the linear equation are randomly selected.
- To demonstrate robustness under FCM assumption violations, we also include confounded versions of ANM-GP and HNM-GP, denoted as ANM-GP-c and HNM-GP-c, respectively:

$$X_i = \text{GP}(Z, \boldsymbol{P}_i; \boldsymbol{\theta}_i) + \text{GMM}(\boldsymbol{\mu}_i, \boldsymbol{\sigma}_i), \quad X_i = \text{GP}(\boldsymbol{P}_i; \boldsymbol{\theta}_{i,1}) + \text{GP}(Z, \boldsymbol{P}_i; \boldsymbol{\theta}_{i,2}) \cdot \text{GMM}(\boldsymbol{\mu}_i, \boldsymbol{\sigma}_i),$$

  where $Z \sim \text{GMM}(\boldsymbol{\mu}_0, \boldsymbol{\sigma}_0)$ is a common confounding variable.

Except for gCastle, other datasets are internally standardized to iSCM (Ormaniec et al., 2025) to eliminate potential sortability. Specifically, during data synthesis, we are required to add an extra step such that $X_i \leftarrow (X_i - \mathbb{E}[X_i])/\sqrt{\text{Var}[X_i]}$ before processing its children.

**Baselines**   In the main text, we list the open-source baselines involved in our experiments. Below are the assumptions underlying these methods, providing context for interpreting the experimental results:

- **Sortability**: Var-Sort (Reisach et al., 2021) and R2-Sort (Reisach et al., 2023), as synthetic SCMs may contain unintentional "fingerprint" information. Ormaniec et al. (2025) proposed a simple method to eliminate such artifacts.
- **Linear Non-Gaussian Acyclic Model**: ICA-LiNGAM (Shimizu et al., 2006) and Direct-LiNGAM (Shimizu et al., 2011) for linear mechanism and non-Gaussian noise.
- **Additive Noise Model**: SCORE (Rolland et al., 2022), based on score function with Gaussian noise. CaPS (Xu et al., 2024), based on score function with strong parent influence (and its criterion is equivalent to FICO according to Stein's identity). RESIT (Peters et al., 2014), based on HSIC test of residuals. NoGAM (Montagna et al., 2023), based on score function of residuals, supporting arbitrary noise.

- **Heteroscedastic Noise Model**: HOST (Duong & Nguyen, 2023), based on Shapiro-Wilk test for gaussian residuals. SKEW (Lin et al., 2025), based on score function with symmetric noise.
- **Minimum Description Length**: Topic (Xu et al., 2025), based on information-geometric theory.

Compared to the above baselines, FICO is characterized by fewer assumptions, being solely related to QPE rather than based on a specific FCM. This broadens its theoretical applicability. However, while FICO has a wider theoretical scope, it does not guarantee superior performance over these models. This is because, even when their specific theoretical assumptions are violated, inherent characteristics within the baselines may still enable them to perform effectively. For instance, Montagna et al. (2023) showed that score function based methods exhibit robustness to assumption violations in a wide range of experiments.

We do not include other algorithms that primarily identify causal graphs, such as NOTEARS (Zheng et al., 2018), Grad-GNN (Yu et al., 2019), and DAGMA Bello et al. (2022), despite their ability to identify unique causal graphs (due to an implicit ANM assumption). These methods mainly output causal graphs, not causal orderings, which preclude accurate calculation of the OD and ODR metrics. Furthermore, this paper focuses exclusively on the causal ordering task. The graph pruning process is omitted because, under consistent pruning strategies, OD and ODR sufficiently reflect the discrepancy between the final DAG and the ground truth.

**Metrics** For causal ordering tasks, we primarily use Order Divergence (Rolland et al., 2022) to measure how well the output causal order aligns with the underlying true DAG. Specifically, for an output causal order $\pi$ and an adjacency matrix $\boldsymbol{A}$ of the underlying true DAG, we define

$$\text{OD}(\pi, \boldsymbol{A}) = \sum_{i=1}^{d} \sum_{\pi_i > \pi_j} \boldsymbol{A}_{i,j}, \qquad \text{ODR}(\pi, \boldsymbol{A}) = \text{OD}(\pi, \boldsymbol{A}) / \sum_{i=1}^{d} \sum_{j=1}^{d} \boldsymbol{A}_{i,j}.$$

In other words, OD reflects how many edges in the DAG violate the causal order $\pi$, and ODR scales OD to the unit interval. When they are both zero, $\pi$ is exactly a topological order of the true DAG.

**Runtime** Regarding hardware, all methods can be run on CPU. Additionally, all score functions are estimated using a kernel-based score matching algorithm from (Rolland et al., 2022). This eliminates the need for extensive GPU training, providing a fast and fair baseline, albeit with some sacrifice in accuracy. For the environment, all methods are based on Python. We run each experiment using the default configurations of these methods. The average time taken for causal ordering is detailed in Table 11.

**Complexity analysis** In our implementation, the primary bottleneck for FICO comes from the score function estimation in each iteration. According to the implementation by (Rolland et al., 2022), assuming $N$ is the sample size and $d$ is the dimension, each estimation step requires $\mathcal{O}(N^3 + N^2 d)$. Since FICO requires $d - 1$ estimations in total, the overall complexity is $\mathcal{O}(N^3 d + N^2 d^2)$. This implies that when $N \gg d$, the sample size becomes the bottleneck, leading to severe inefficiency. For large samples, denoising score matching (Sanchez et al., 2023) can be applied, which amortize the complexity over multiple steps, resulting in a single-step optimization complexity of only $\mathcal{O}(N \max(d, c))$, where $c$ is the number of model parameters.

# D   ADDITIONAL RESULTS

## D.1   HYPER-PARAMETER TUNING FOR QPE-F

**Hyperparameters for QPE-f**   A primary source of hyperparameters for QPE-f arises from the necessity to initially fit a causal flow (Equation 1). Specifically, we employ a discrete causal flow, where the causal flow $u_\theta$ is parameterized as a composition of several invertible transformations:

$$T_{1,\theta} \circ T_{2,\theta} \circ \cdots \circ T_{t,\theta},$$

where each invertible transformation $T_{i,\theta}(\boldsymbol{x}, y)$ is a monotonic function w.r.t. $y$. Candidate choices for these transformations include affine transformations (Dinh et al., 2017), Rational-Quadratic Spline (RQS) transformations (Durkan et al., 2019), Monotonic Neural Networks (MNN) (Huang et al., 2018), and Unconstrained Monotonic Neural Networks (UMNN) (Wehenkel & Louppe, 2019). These methods exhibit progressively increasing representational capacity. In our experimental setup, affine transformations were excluded as they restrict the model to representing only HNM (Khemakhem et al., 2021), thereby failing to generalize to broader model classes.

The second source of hyperparameters for QPE-f pertains to the selection of the hypothesized QPE model employed for the linear span test. We consider the following choices for the hypothesized model: **(i)** Hypernetwork: an unconstrained neural network that implicitly satisfies the hypothesis; **(ii)** Low-rank network: $\sum_{i=1}^{K} a_i(\boldsymbol{x}; \theta_i)\, b_i(y; \phi_i)$, where both $a_i$ and $b_i$ are unconstrained neural networks, explicitly representing a finite number of basis functions in the hypothesis; **(iii)** Polynomial network: $\sum_{i=0}^{K} a_i(\boldsymbol{x}; \theta_i)\, y^K$, explicitly assuming polynomial basis functions.

For training both the flow and the hypothesized QPE models, the Adam optimizer was employed with a learning rate of 0.01 and a weight decay of 0.001. To ensure continuity, networks utilized the SiLU activation function and comprised two hidden layers, each with a width of 100. Training proceeded for a total of 1000 epochs, and the model corresponding to the epoch with the minimum loss function value was selected for basis test. The training and validation sets were not partitioned.

**Hyperparameter tuning**   We conducted hyperparameter tuning across several dimensions: the choice of transformation type (RQS, MNN, UMNN), the number of composite transformations ($t \in \{1, 2, 5\}$), the hypothesized QPE model (Hypernetwork, Low-rank network, Polynomial network), and the specific hyperparameters for the hypothesized QPE models (e.g., rank for Low-rank networks, degree for Polynomial networks). Due to space constraints, the hyperparameter tuning results for the SIM and Tue datasets are presented in Tables 4 and 5, respectively.

Table 4: Hyperparameter tuning on the SIM dataset. Best per flow configuration is bolded.

| Test | $K$ | RQS $t=1$ | RQS $t=2$ | RQS $t=5$ | MNN $t=1$ | MNN $t=2$ | MNN $t=5$ | UMNN $t=1$ | UMNN $t=2$ | UMNN $t=5$ |
|---|---|---|---|---|---|---|---|---|---|---|
| Hyper | - | 0.75 (0.76) | 0.67 (0.60) | 0.57 (0.52) | 0.83 (0.83) | 0.75 (0.77) | 0.78 (0.82) | 0.79 (0.78) | 0.76 (0.75) | 0.67 (0.68) |
| Lowrank | 2 | 0.76 (0.77) | 0.68 (0.60) | 0.57 (0.53) | **0.87** (0.87) | 0.75 (0.75) | 0.75 (0.76) | 0.79 (0.80) | 0.79 (0.81) | 0.73 (0.73) |
| | 5 | 0.76 (0.77) | 0.64 (0.58) | 0.54 (0.47) | 0.80 (0.81) | 0.78 (0.78) | 0.73 (0.74) | 0.74 (0.72) | 0.83 (0.82) | 0.73 (0.74) |
| | 10 | 0.75 (0.76) | 0.64 (0.56) | 0.54 (0.48) | 0.81 (0.81) | 0.79 (0.79) | 0.73 (0.75) | 0.77 (0.73) | 0.76 (0.76) | 0.72 (0.72) |
| Poly | 0 | **0.87** (0.88) | **0.82** (0.76) | **0.76** (0.74) | 0.83 (0.83) | **0.85** (0.85) | **0.85** (0.85) | **0.88** (0.86) | **0.87** (0.87) | 0.73 (0.77) |
| | 1 | 0.84 (0.83) | 0.78 (0.75) | 0.69 (0.68) | 0.83 (0.82) | 0.79 (0.80) | 0.79 (0.84) | 0.85 (0.83) | 0.86 (0.85) | **0.75** (0.77) |
| | 2 | 0.81 (0.79) | 0.73 (0.70) | 0.69 (0.69) | 0.81 (0.81) | 0.74 (0.76) | 0.78 (0.83) | 0.84 (0.83) | 0.81 (0.82) | 0.74 (0.73) |

Table 5: Hyperparameter tuning on the Tue dataset. Best per flow configuration is bolded.

| Test | $K$ | RQS $t=1$ | RQS $t=2$ | RQS $t=5$ | MNN $t=1$ | MNN $t=2$ | MNN $t=5$ | UMNN $t=1$ | UMNN $t=2$ | UMNN $t=5$ |
|---|---|---|---|---|---|---|---|---|---|---|
| Hyper | - | 0.46 (0.41) | 0.47 (0.49) | 0.56 (0.48) | 0.59 (0.61) | 0.59 (0.53) | 0.49 (0.51) | 0.58 (0.55) | 0.48 (0.45) | 0.51 (0.45) |
| Lowrank | 2 | 0.43 (0.36) | 0.52 (0.52) | 0.52 (0.47) | 0.61 (0.61) | 0.56 (0.54) | 0.51 (0.46) | 0.53 (0.48) | 0.43 (0.46) | 0.52 (0.45) |
| | 5 | 0.47 (0.37) | 0.53 (0.55) | 0.57 (0.45) | 0.66 (0.67) | 0.53 (0.52) | 0.49 (0.47) | 0.56 (0.57) | 0.42 (0.45) | 0.51 (0.45) |
| | 10 | 0.47 (0.37) | 0.51 (0.52) | 0.56 (0.46) | 0.67 (0.66) | **0.63** (0.65) | 0.48 (0.44) | 0.56 (0.56) | 0.42 (0.46) | 0.48 (0.43) |
| Poly | 0 | 0.52 (0.45) | **0.56** (0.51) | **0.60** (0.49) | 0.56 (0.57) | 0.56 (0.55) | **0.58** (0.62) | 0.46 (0.41) | 0.45 (0.35) | 0.47 (0.46) |
| | 1 | 0.48 (0.40) | **0.56** (0.53) | **0.60** (0.53) | 0.62 (0.60) | 0.59 (0.57) | 0.53 (0.49) | 0.57 (0.52) | 0.56 (0.49) | 0.54 (0.42) |
| | 2 | **0.55** (0.52) | 0.53 (0.51) | 0.58 (0.52) | **0.70** (0.78) | 0.61 (0.61) | 0.49 (0.49) | **0.64** (0.57) | **0.58** (0.60) | **0.58** (0.52) |

**Optimal hyperparameter configurations**  Table 6 provides the empirically determined optimal hyperparameter configurations for each dataset, thereby ensuring experimental reproducibility.

Table 6: Empirically optimal hyperparameter configurations for each dataset.

| Dataset | Transform | $t$ | Test | $K$ | Dataset | Transform | $t$ | Test | $K$ | Dataset | Transform | $t$ | Test | $K$ |
|---|---|---|---|---|---|---|---|---|---|---|---|---|---|---|
| AN | UMNN | 1 | Poly | 0 | SIM-ln | MNN | 2 | Poly | 0 | D4-s2c | MNN | 1 | Lowrank | 10 |
| AN-s | MNN | 1 | Poly | 0 | Cha | MNN | 1 | Lowrank | 10 | Per | RQS | 1 | Hyper | |
| LS | UMNN | 2 | Poly | 0 | Net | RQS | 2 | Poly | 0 | Sig | RQS | 1 | Poly | 1 |
| LS-s | UMNN | 1 | Poly | 1 | Multi | UMNN | 2 | Poly | 1 | Vex | UMNN | 1 | Lowrank | 10 |
| MNU | RQS | 1 | Poly | 1 | Tue | MNN | 1 | Poly | 2 | Qd-V | MNN | 1 | Poly | 1 |
| SIM | UMNN | 1 | Poly | 0 | D4-s1 | UMNN | 5 | Lowrank | 5 | Sig-V | UMNN | 2 | Poly | 2 |
| SIM-c | MNN | 1 | Poly | 0 | D4-s2a | MNN | 2 | Poly | 2 | Rbf-V | UMNN | 2 | Poly | 0 |
| SIM-g | MNN | 2 | Poly | 0 | D4-s2b | MNN | 1 | Poly | 2 | NN-V | MNN | 1 | Lowrank | 10 |

## D.2 COMPARATIVE EVALUATION OF QPE-K, QPE-F, AND BASELINES

Tables 7 to 9 present the comprehensive results of 20 distinct methods across 24 datasets. In this extensive comparison, QPE-f consistently achieves state-of-the-art performance across the vast majority of datasets, with only minor exceptions where it is marginally outperformed by certain methods. A direct comparison between QPE-f and CVEL, as well as QPE-k, empirically validates the assertion made in Section 4.3 regarding the superior accuracy of flow-based methods in QPE estimation. Furthermore, CVEL and QPE-f consistently outperform other methods in both non-ANM and non-HNM datasets (specifically, flow synthesized Per, Sig, Vex; and constraint QPE synthesized Qd-V, Sig-V, Rbf-V, NN-V). This empirically corroborates the theoretical insight that the identifiability conditions for causal velocity or QPE are more broadly applicable than those only for ANM and HNM.

## D.3 CONVERGENCE OF FICO

Figure 4 illustrates the convergence behavior of 3 score-based methods (SKEW, SCORE, and FICO) w.r.t. sample size on HNM-GP datasets. All methods utilize the same score function estimation algorithm, which is known to asymptotically approach the true score function with increasing sample size. Figure 4 specifically demonstrates how the accuracy of these methods in the causal ordering evolves as the score function estimates become increasingly precise.

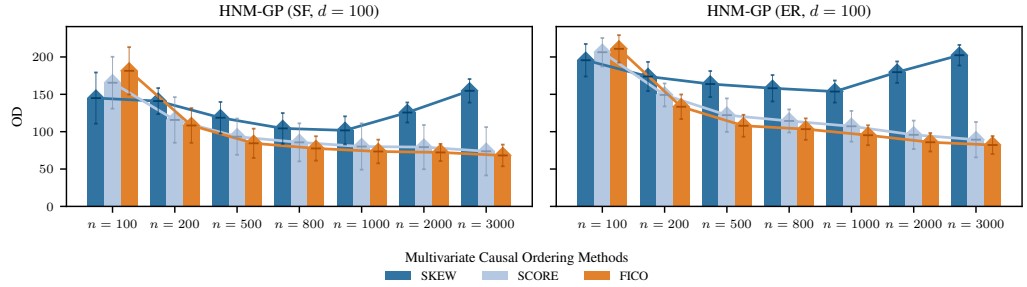

Figure 4: Convergence behavior of SKEW, SCORE, and FICO on HNM-GP datasets.

Notably, the SKEW method exhibits a degradation in performance as the score function estimate improves, primarily because its underlying assumptions (e.g., symmetric noise) are violated in this context. In contrast, SCORE and FICO maintain consistent convergence profiles, likely attributable to the satisfaction of their intrinsic assumptions or properties. Furthermore, FICO consistently outperforms SCORE as the sample size gradually increases.

Table 7: Accuracy (and AUDRC) of QPE-k, QPE-f, and 21 baselines on 9 bivariate datasets.

| Method | AN | AN-s | LS | LS-s | MNU | SIM | SIM-c | SIM-g | SIM-ln |
|---|---|---|---|---|---|---|---|---|---|
| ICA-LiNGAM | 0.63 (0.61) | 0.63 (0.61) | 0.63 (0.61) | 0.63 (0.61) | 0.63 (0.61) | 0.63 (0.61) | 0.63 (0.61) | 0.63 (0.61) | 0.63 (0.61) |
| VAR-LiNGAM | 0.06 (0.07) | 0.11 (0.11) | 0.00 (0.00) | 0.00 (0.00) | 0.00 (0.00) | 0.42 (0.36) | 0.47 (0.48) | 0.27 (0.25) | 0.23 (0.14) |
| Direct-LiNGAM | 0.06 (0.07) | 0.00 (0.00) | 0.11 (0.11) | 0.00 (0.00) | 0.00 (0.00) | 0.42 (0.36) | 0.47 (0.48) | 0.26 (0.25) | 0.23 (0.14) |
| ANM | 0.43 (0.35) | 0.47 (0.42) | 0.46 (0.50) | 0.45 (0.47) | 0.40 (0.37) | 0.45 (0.55) | 0.49 (0.48) | 0.41 (0.42) | 0.46 (0.48) |
| CAM | **1.00** (1.00) | **1.00** (1.00) | **1.00** (1.00) | 0.51 (0.50) | 0.91 (0.93) | 0.59 (0.58) | 0.59 (0.53) | 0.80 (0.79) | 0.88 (0.90) |
| RESIT | 0.99 (1.00) | **1.00** (1.00) | 0.72 (0.76) | 0.09 (0.07) | 0.01 (0.00) | 0.78 (0.78) | 0.82 (0.85) | 0.76 (0.72) | 0.67 (0.60) |
| RECI | 0.18 (0.27) | 0.35 (0.34) | 0.22 (0.16) | 0.44 (0.48) | 0.13 (0.16) | 0.44 (0.40) | 0.53 (0.63) | 0.39 (0.30) | 0.44 (0.40) |
| CGNN | 0.96 (0.94) | 0.57 (0.57) | 0.92 (0.88) | 0.64 (0.64) | 0.94 (0.95) | 0.75 (0.76) | 0.76 (0.78) | 0.72 (0.60) | 0.75 (0.67) |
| CDS | 0.99 (0.98) | 0.99 (1.00) | 0.76 (0.79) | 0.05 (0.06) | 0.70 (0.74) | 0.71 (0.65) | 0.76 (0.82) | 0.73 (0.76) | 0.65 (0.63) |
| PNL | 0.30 (0.31) | 0.49 (0.48) | 0.33 (0.35) | 0.49 (0.53) | 0.58 (0.58) | 0.46 (0.46) | 0.54 (0.50) | 0.43 (0.46) | 0.42 (0.37) |
| QCCD | **1.00** (1.00) | 0.83 (0.75) | **1.00** (1.00) | **1.00** (1.00) | **1.00** (1.00) | 0.68 (0.69) | 0.77 (0.75) | 0.67 (0.63) | 0.87 (0.87) |
| CAREFL | **1.00** (1.00) | **1.00** (1.00) | **1.00** (1.00) | **1.00** (1.00) | **1.00** (1.00) | 0.79 (0.81) | 0.83 (0.87) | 0.78 (0.72) | 0.82 (0.85) |
| HECI | 0.98 (0.97) | 0.55 (0.60) | 0.92 (0.86) | 0.55 (0.60) | 0.33 (0.36) | 0.49 (0.42) | 0.55 (0.64) | 0.56 (0.52) | 0.65 (0.60) |
| GRCI | 0.67 (0.64) | 0.68 (0.60) | 0.64 (0.65) | 0.44 (0.45) | 0.47 (0.49) | 0.55 (0.59) | 0.65 (0.67) | 0.41 (0.45) | 0.52 (0.42) |
| LOCI | **1.00** (1.00) | **1.00** (1.00) | **1.00** (1.00) | **1.00** (1.00) | **1.00** (1.00) | 0.78 (0.80) | 0.81 (0.87) | 0.78 (0.77) | 0.80 (0.76) |
| CDCI | **1.00** (1.00) | 0.96 (0.98) | **1.00** (1.00) | 0.97 (0.97) | 0.99 (0.99) | 0.84 (0.83) | 0.76 (0.83) | 0.73 (0.63) | 0.79 (0.82) |
| IGCI | 0.89 (0.83) | 0.97 (0.99) | 0.95 (0.95) | 0.94 (0.91) | 0.86 (0.85) | 0.36 (0.36) | 0.42 (0.37) | 0.86 (0.88) | 0.59 (0.54) |
| SLOPE | 0.14 (0.23) | 0.26 (0.21) | 0.16 (0.11) | 0.15 (0.20) | 0.03 (0.06) | 0.43 (0.40) | 0.52 (0.63) | 0.45 (0.36) | 0.44 (0.34) |
| SLOPPY | **1.00** (1.00) | **1.00** (1.00) | **1.00** (1.00) | 0.56 (0.50) | 0.96 (0.97) | 0.64 (0.70) | 0.62 (0.61) | 0.82 (0.80) | 0.86 (0.90) |
| DIVOT | 0.62 (0.58) | 0.69 (0.73) | 0.45 (0.47) | 0.69 (0.68) | **1.00** (1.00) | 0.68 (0.72) | 0.47 (0.41) | 0.60 (0.61) | 0.63 (0.68) |
| CVEL | **1.00** (1.00) | 0.98 (1.00) | 0.98 (0.97) | 0.93 (0.91) | 0.94 (0.93) | 0.63 (0.66) | 0.72 (0.73) | **0.90** (0.87) | 0.76 (0.73) |
| QPE-k | 0.99 (0.96) | 0.88 (0.75) | **1.00** (1.00) | 0.78 (0.80) | **1.00** (1.00) | 0.83 (0.85) | 0.79 (0.83) | 0.83 (0.81) | 0.68 (0.69) |
| QPE-f | **1.00** (1.00) | **1.00** (1.00) | **1.00** (1.00) | 0.99 (1.00) | **1.00** (1.00) | **0.88** (0.86) | **0.88** (0.92) | 0.86 (0.84) | **0.92** (0.91) |

Table 8: Accuracy (and AUDRC) of QPE-k, QPE-f, and 21 baselines on 8 bivariate datasets.

| Method | Cha | Net | Multi | Tue | D4-s1 | D4-s2a | D4-s2b | D4-s2c |
|---|---|---|---|---|---|---|---|---|
| ICA-LiNGAM | 0.52 (0.57) | 0.52 (0.57) | 0.52 (0.57) | 0.64 (0.61) | 0.67 (0.61) | 0.50 (0.56) | 0.52 (0.55) | 0.51 (0.56) |
| VAR-LiNGAM | 0.55 (0.54) | 0.31 (0.32) | 0.34 (0.40) | 0.56 (0.48) | 0.58 (0.63) | 0.53 (0.51) | 0.55 (0.54) | 0.57 (0.62) |
| Direct-LiNGAM | 0.54 (0.54) | 0.31 (0.32) | 0.35 (0.41) | 0.51 (0.46) | 0.67 (0.72) | 0.61 (0.59) | 0.59 (0.52) | 0.62 (0.55) |
| ANM | 0.41 (0.37) | 0.47 (0.46) | 0.48 (0.42) | 0.65 (0.67) | 0.50 (0.70) | 0.48 (0.52) | 0.46 (0.46) | 0.48 (0.47) |
| CAM | 0.48 (0.48) | 0.78 (0.81) | 0.35 (0.36) | 0.55 (0.54) | 0.42 (0.57) | 0.35 (0.31) | 0.44 (0.39) | 0.44 (0.49) |
| RESIT | 0.74 (0.79) | 0.76 (0.80) | 0.37 (0.43) | 0.63 (0.61) | 0.58 (0.72) | 0.63 (0.65) | 0.55 (0.53) | 0.54 (0.49) |
| RECI | 0.56 (0.59) | 0.60 (0.62) | 0.85 (0.85) | 0.64 (0.55) | 0.58 (0.64) | 0.58 (0.69) | 0.50 (0.64) | 0.52 (0.55) |
| CGNN | 0.61 (0.66) | 0.75 (0.76) | 0.84 (0.83) | 0.69 (0.68) | 0.50 (0.63) | 0.59 (0.59) | 0.47 (0.57) | 0.50 (0.54) |
| CDS | 0.71 (0.77) | 0.78 (0.80) | 0.44 (0.46) | 0.67 (0.69) | 0.58 (0.70) | 0.59 (0.60) | 0.54 (0.53) | 0.58 (0.57) |
| PNL | 0.45 (0.36) | 0.51 (0.47) | 0.45 (0.41) | 0.51 (0.54) | 0.33 (0.39) | 0.45 (0.32) | 0.59 (0.63) | 0.39 (0.40) |
| QCCD | 0.55 (0.54) | 0.81 (0.85) | 0.49 (0.49) | 0.68 (0.73) | 0.33 (0.52) | 0.55 (0.53) | 0.53 (0.41) | 0.53 (0.53) |
| CAREFL | 0.72 (0.76) | 0.85 (0.83) | 0.76 (0.79) | 0.63 (0.60) | 0.58 (0.64) | 0.69 (0.69) | **0.63** (0.68) | 0.56 (0.50) |
| HECI | 0.57 (0.59) | 0.72 (0.71) | 0.91 (0.90) | 0.61 (0.57) | 0.42 (0.62) | 0.56 (0.66) | 0.50 (0.63) | 0.47 (0.50) |
| GRCI | 0.53 (0.55) | 0.58 (0.53) | 0.61 (0.58) | 0.55 (0.56) | 0.67 (0.79) | 0.51 (0.57) | 0.61 (0.70) | 0.50 (0.54) |
| LOCI | 0.73 (0.77) | 0.87 (0.88) | 0.79 (0.80) | 0.61 (0.67) | 0.58 (0.55) | 0.69 (0.74) | 0.61 (0.66) | 0.54 (0.47) |
| CDCI | 0.67 (0.71) | 0.84 (0.80) | 0.92 (0.93) | 0.68 (0.78) | 0.67 (0.69) | 0.68 (0.65) | 0.60 (0.59) | 0.61 (0.55) |
| IGCI | 0.55 (0.54) | 0.57 (0.58) | 0.68 (0.66) | 0.62 (0.65) | 0.42 (0.25) | 0.44 (0.44) | 0.43 (0.42) | 0.40 (0.44) |
| SLOPE | 0.56 (0.59) | 0.61 (0.60) | 0.88 (0.88) | 0.57 (0.55) | 0.33 (0.52) | 0.51 (0.59) | 0.40 (0.52) | 0.39 (0.43) |
| SLOPPY | 0.48 (0.49) | 0.80 (0.82) | 0.46 (0.47) | 0.64 (0.63) | 0.33 (0.48) | 0.33 (0.28) | 0.44 (0.40) | 0.44 (0.49) |
| DIVOT | 0.44 (0.38) | 0.49 (0.51) | 0.34 (0.38) | 0.38 (0.42) | 0.50 (0.54) | 0.57 (0.52) | 0.55 (0.49) | 0.55 (0.51) |
| CVEL | 0.68 (0.73) | 0.62 (0.60) | **0.97** (0.96) | 0.64 (0.59) | 0.67 (0.70) | 0.51 (0.50) | 0.58 (0.53) | 0.58 (0.57) |
| QPE-k | 0.60 (0.63) | **0.89** (0.90) | 0.88 (0.89) | 0.54 (0.47) | 0.58 (0.75) | 0.67 (0.69) | 0.61 (0.64) | **0.64** (0.61) |
| QPE-f | **0.85** (0.87) | 0.86 (0.87) | 0.96 (0.96) | **0.70** (0.78) | **0.79** (0.78) | **0.71** (0.72) | 0.62 (0.66) | 0.60 (0.48) |

Table 9: Accuracy (and AUDRC) of QPE-k, QPE-f, and 21 baselines on 7 bivariate datasets.

| Method | Per | Sig | Vex | Qd-V | Sig-V | Rbf-V | NN-V |
|---|---|---|---|---|---|---|---|
| ICA-LiNGAM | 0.63 (0.61) | 0.63 (0.61) | 0.63 (0.61) | 0.63 (0.61) | 0.63 (0.61) | 0.63 (0.61) | 0.63 (0.61) |
| VAR-LiNGAM | 0.67 (0.63) | 0.36 (0.36) | 0.48 (0.49) | 0.87 (0.85) | 0.59 (0.66) | 0.61 (0.55) | 0.65 (0.69) |
| Direct-LiNGAM | 0.67 (0.63) | 0.37 (0.36) | 0.47 (0.47) | 0.88 (0.89) | 0.59 (0.66) | 0.60 (0.54) | 0.66 (0.69) |
| ANM | 0.49 (0.51) | 0.44 (0.50) | 0.39 (0.39) | 0.49 (0.57) | 0.50 (0.50) | 0.43 (0.51) | 0.48 (0.50) |
| CAM | 0.00 (0.00) | 0.09 (0.05) | 0.24 (0.21) | 0.12 (0.16) | 0.47 (0.48) | 0.30 (0.22) | 0.23 (0.22) |
| RESIT | 0.70 (0.72) | 0.20 (0.22) | 0.03 (0.03) | 0.80 (0.82) | 0.75 (0.75) | 0.54 (0.49) | 0.61 (0.72) |
| RECI | 0.00 (0.00) | 0.07 (0.06) | 0.94 (0.94) | 0.63 (0.59) | 0.53 (0.52) | 0.19 (0.20) | 0.49 (0.52) |
| CGNN | 0.82 (0.80) | 0.68 (0.66) | 0.77 (0.77) | 0.71 (0.79) | 0.76 (0.77) | 0.68 (0.67) | 0.64 (0.73) |
| CDS | 0.18 (0.17) | 0.08 (0.09) | 0.04 (0.07) | 0.78 (0.76) | 0.66 (0.66) | 0.45 (0.43) | 0.52 (0.54) |
| PNL | 0.42 (0.40) | 0.43 (0.48) | 0.38 (0.37) | 0.46 (0.44) | 0.43 (0.45) | 0.51 (0.53) | 0.41 (0.37) |
| QCCD | 0.02 (0.01) | 0.14 (0.12) | 0.04 (0.03) | 0.34 (0.29) | 0.55 (0.56) | 0.32 (0.26) | 0.33 (0.35) |
| CAREFL | 0.95 (0.97) | 0.64 (0.65) | 0.91 (0.92) | 0.72 (0.73) | 0.91 (0.89) | 0.61 (0.53) | 0.84 (0.82) |
| HECI | 0.01 (0.00) | 0.13 (0.15) | 0.94 (0.94) | 0.59 (0.53) | 0.53 (0.48) | 0.19 (0.20) | 0.45 (0.51) |
| GRCI | 0.56 (0.48) | 0.54 (0.54) | 0.54 (0.55) | 0.47 (0.40) | 0.51 (0.52) | 0.47 (0.45) | 0.60 (0.55) |
| LOCI | 0.96 (0.97) | 0.70 (0.69) | 0.87 (0.86) | 0.71 (0.72) | 0.87 (0.88) | 0.61 (0.55) | 0.78 (0.79) |
| CDCI | 0.48 (0.47) | 0.42 (0.45) | 0.49 (0.52) | 0.74 (0.75) | 0.80 (0.77) | 0.57 (0.58) | 0.72 (0.72) |
| IGCI | **1.00** (1.00) | 0.77 (0.81) | 0.87 (0.89) | 0.47 (0.54) | 0.49 (0.52) | 0.58 (0.65) | 0.48 (0.53) |
| SLOPE | 0.00 (0.00) | 0.06 (0.06) | 0.93 (0.93) | 0.61 (0.57) | 0.52 (0.46) | 0.18 (0.19) | 0.44 (0.45) |
| SLOPPY | 0.02 (0.02) | 0.11 (0.08) | 0.10 (0.07) | 0.17 (0.14) | 0.48 (0.49) | 0.32 (0.26) | 0.33 (0.34) |
| DIVOT | 0.97 (0.96) | 0.82 (0.84) | 0.05 (0.04) | 0.32 (0.33) | 0.44 (0.49) | 0.63 (0.58) | 0.47 (0.44) |
| CVEL | **1.00** (1.00) | 0.84 (0.81) | **0.96** (0.97) | **0.91** (0.92) | **0.94** (0.93) | 0.92 (0.92) | 0.87 (0.92) |
| QPE-k | 0.77 (0.79) | 0.89 (0.84) | 0.63 (0.60) | 0.42 (0.44) | 0.67 (0.73) | 0.68 (0.66) | 0.53 (0.49) |
| QPE-f | **1.00** (1.00) | **0.90** (0.93) | 0.91 (0.92) | **0.91** (0.91) | 0.91 (0.90) | **0.94** (0.94) | **0.90** (0.93) |

### D.4 FICO'S PERFORMANCE UNDER HETEROSCEDASTIC GAUSSIAN ASSUMPTION

We analyze the interpretability of FICO under the heteroscedastic Gaussian assumption concerning the mean and variance functions in Appendix B.2. Figure 5 details this relationship. The synthetic dataset with heteroscedastic Gaussian noise is generated by $X_i = \mu_i(\boldsymbol{P}_i) + \sigma_i(\boldsymbol{P}_i)\mathcal{N}(0,1)$, where $\mu_i = h(\cdot; \boldsymbol{a}_i, b_i, \alpha, 1)$ and $\sigma_i = \log(\exp(h(\cdot; \boldsymbol{c}_i, d_i, \alpha, \beta)) + 1)$. Here, $h(\cdot; \boldsymbol{a}_i, b_i, \alpha, \beta)$ is an affine function with sin perturbations:

$$\beta\left(\sum_{P_j \in \boldsymbol{P}_i} a_{i,j} P_j + b_i\right) + \alpha\left(\sum_{P_j \in \boldsymbol{P}_i} \sin(P_{i,j})\right),$$

where coefficients $a_{i,j}, c_{i,j} \sim \mathcal{N}(0,1)$ and biases $b_i, d_i \sim \mathcal{N}(0,2)$. Hyperparameters $\alpha$ and $\beta$ control the magnitude of the sin perturbation and the gradient of the affine function, respectively. Indirectly, $\beta$ controls the magnitude of $|\kappa| = |\sigma_i'/\mu_i'|$ (since $\beta$ for $\mu_i$ is always 1), while $\alpha$ controls how closely $\mu_i$ and $\sigma_i$ approximate linear functions. Each cell in Figure 5 shows the average over 100 sub-tests, where corresponding sub-tests share the same coefficients and biases, varying only hyperparameters $\alpha$ and $\beta$.

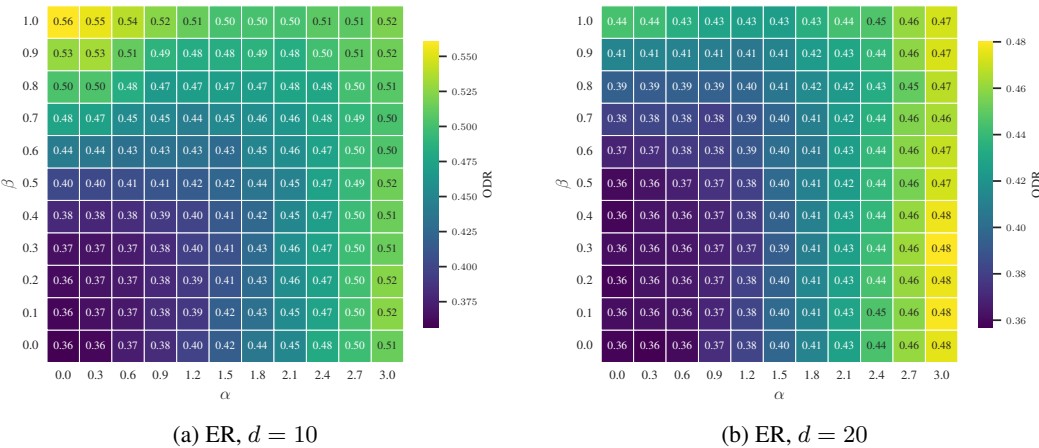

(a) ER, $d = 10$          (b) ER, $d = 20$

Figure 5: Relationship between FICO's ODR and hyperparameters $\alpha$ and $\beta$ under the heteroscedastic Gaussian assumption. **(a)** 10-variable ER graph; **(b)** 20-variable ER graph. The expected numbers of edges in these graphs is 4 times their dimensions.

The results indicate that FICO's performance gradually degrades as $\alpha$ and $\beta$ increase. This reflects that the magnitude of $|\kappa|$ and the linearity of $\mu_i$ and $\sigma_i$ indeed affect the validity of Assumption 5.4. FICO performs best when $|\kappa|$ is sufficiently small and both $\mu_i$ and $\sigma_i$ are linear. As the assumption is progressively violated, the performance weakens, which is empirically consistent with our analysis.

### D.5 FICO ON REAL-WORLD DATASETS

As shown in Table 10, on real-world datasets, score function based methods generally perform poorly on Sachs, likely due to the assumption not holding. However, CaPS and FICO achieve the best performance on Syntren.

### D.6 FICO ON SYNTHETIC DATASETS

Figure 6 presents the experimental results across 8 synthetic datasets under 2 types of random graphs (SF and ER). It is important to note that ODR exhibits a hidden baseline at 0.5, which corresponds to random ordering (since the probability of $\pi_i > \pi_j$ is 0.5 in a

Table 10: ODR on real-world datasets for different methods. The best is bolded.

| Method | Sachs | Syntren |
|---|---|---|
| R2-Sort | 0.29 | $0.89 \pm 0.07$ |
| Var-Sort | 0.82 | $0.50 \pm 0.17$ |
| ICA-LiNGAM | 0.59 | $0.42 \pm 0.19$ |
| Direct-LiNGAM | 0.47 | $0.62 \pm 0.10$ |
| HOST | **0.18** | $0.42 \pm 0.10$ |
| RESIT | 0.47 | $0.74 \pm 0.14$ |
| TOPIC | 0.59 | $0.38 \pm 0.13$ |
| NoGAM | 0.65 | $0.39 \pm 0.08$ |
| SKEW | 0.71 | $0.49 \pm 0.15$ |
| SCORE | 0.71 | $0.38 \pm 0.10$ |
| CaPS | 0.71 | $\mathbf{0.33 \pm 0.08}$ |
| FICO | 0.71 | $\mathbf{0.33 \pm 0.08}$ |

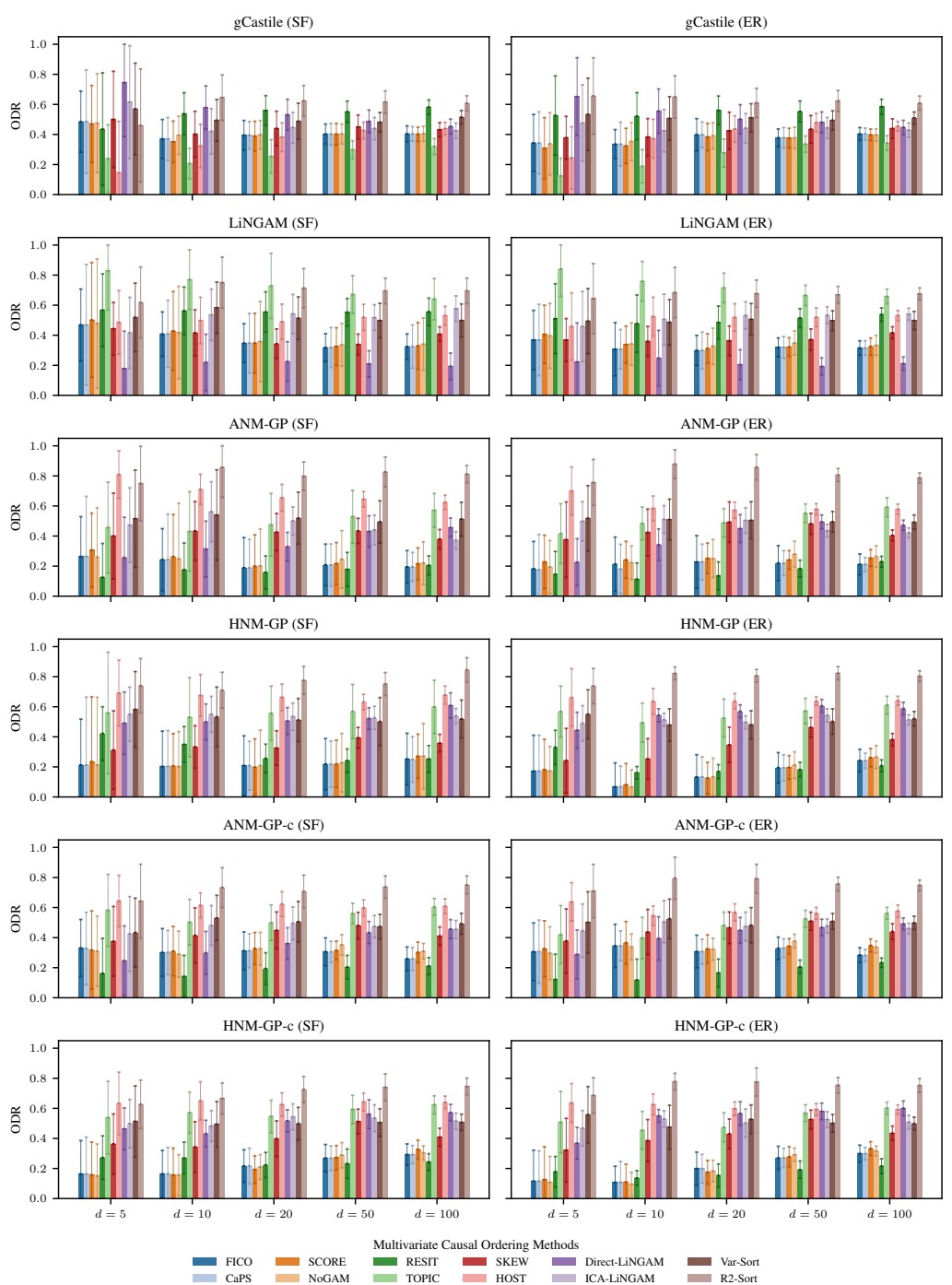

Figure 6: ODRs of FICO and baselines on 12 multivariate causal discovery datasets. Lower is better.

random ordering). An ODR value less than 0.5 indicates that the method effectively retains more than half of the edges, implying that the underlying assumptions or characteristics are met within the dataset. Conversely, an ODR greater than 0.5 suggests that these assumptions or characteristics are likely to be violated. When ODR approaches 0.5, the method essentially performs at a level equivalent to random ordering.

The results presented in Figure 6 reveal several key observations:

- The elimination of "fingerprints" by iSCM leads to ODR values exceeding 0.5 for both Var-Sort and R2-Sort, indicating performance worse than random ordering. This suggests low sortability in these datasets, thereby mitigating the possibility of "hacking" through these specific metrics.
- ICA-LiNGAM and Direct-LiNGAM perform well exclusively in LiNGAM datasets, approximating random ordering in other contexts.
- RESIT exhibits unexpectedly strong performance on GP datasets. This could be due to GMM noise, particularly in high-dimensional settings. This phenomenon warrants further investigation. Conversely, its performance in gCastle and LiNGAM datasets is moderate, nearing random ordering, potentially due to the Gaussian and Gumbel distributions.
- HOST demonstrates significantly superior performance in low-dimensional gCastle scenarios, which may also explain its best performance on Sachs. However, in other contexts, its ODR consistently remains above 0.5, violating assumptions, largely because its normality check on noise becomes progressively challenging in higher dimensions.
- TOPIC exhibits variability. It outperforms other methods in gCastle. Yet, its performance is suboptimal elsewhere. For instance, it is notably above 0.5 in LiNGAM and approximates random ordering in other datasets.
- SKEW's performance is generally slightly lower than other score function based methods due to the violation of its symmetry noise assumption in most datasets, with the exception of the Gaussian noise in gCastle.
- The remaining score function based methods exhibit comparable performance. In high-dimensional settings, CaPS and FICO are marginally superior to SCORE and NoGAM (with a potential difference of $\leq 0.05$). Given their formal equivalence, any performance disparities between CaPS and FICO are solely attributable to precision differences.

Overall, score function based methods demonstrate the robustness reported in Montagna et al. (2023). Despite potentially being slightly outperformed by other methods on specific datasets, their ODR values consistently remain below 0.5 across all experimental settings. This empirically suggests that the underlying assumptions or characteristics of these methods are implicitly satisfied.

## D.7 FICO'S RUNTIME EFFICIENCY

Table 11 presents FICO's runtime performance in comparison to other baseline methods.

Table 11: Runtime efficiency of FICO and baselines, in seconds per sub-test.

| Method | $d = 5$ | $d = 10$ | $d = 20$ | $d = 50$ | $d = 100$ |
|---|---|---|---|---|---|
| R2-Sort | $0.000 \pm 0.000$ | $0.000 \pm 0.000$ | $0.000 \pm 0.000$ | $0.000 \pm 0.000$ | $0.005 \pm 0.007$ |
| Var-Sort | $0.000 \pm 0.000$ | $0.000 \pm 0.000$ | $0.000 \pm 0.000$ | $0.000 \pm 0.000$ | $0.000 \pm 0.000$ |
| ICA-LiNGAM | $0.010 \pm 0.012$ | $0.043 \pm 0.035$ | $0.128 \pm 0.073$ | $0.638 \pm 0.158$ | $2.200 \pm 0.636$ |
| Direct-LiNGAM | $0.010 \pm 0.000$ | $0.050 \pm 0.000$ | $0.338 \pm 0.000$ | $5.076 \pm 0.065$ | $39.942 \pm 0.507$ |
| HOST | $0.625 \pm 0.097$ | $1.967 \pm 0.106$ | $4.159 \pm 0.207$ | $7.177 \pm 0.330$ | $17.332 \pm 0.640$ |
| TOPIC | $0.318 \pm 0.066$ | $1.984 \pm 0.541$ | $9.381 \pm 2.604$ | $60.869 \pm 13.916$ | $250.343 \pm 45.157$ |
| RESIT | $0.268 \pm 0.011$ | $1.091 \pm 0.051$ | $4.772 \pm 0.268$ | $40.010 \pm 2.026$ | $227.662 \pm 9.178$ |
| NoGAM | $1.243 \pm 0.299$ | $4.251 \pm 0.593$ | $14.864 \pm 1.028$ | $92.578 \pm 6.251$ | $370.316 \pm 16.281$ |
| SKEW | $0.543 \pm 0.362$ | $1.202 \pm 0.624$ | $2.548 \pm 0.876$ | $8.022 \pm 1.248$ | $20.043 \pm 1.136$ |
| SCORE | $0.831 \pm 0.462$ | $1.883 \pm 0.737$ | $4.306 \pm 0.982$ | $14.180 \pm 1.914$ | $37.609 \pm 2.007$ |
| CaPS | $0.455 \pm 0.037$ | $1.074 \pm 0.056$ | $2.761 \pm 0.285$ | $10.822 \pm 1.037$ | $33.794 \pm 3.501$ |
| FICO | $0.425 \pm 0.322$ | $0.797 \pm 0.364$ | $1.727 \pm 0.523$ | $5.550 \pm 0.943$ | $13.538 \pm 1.248$ |

Among the methods evaluated, excluding those based on sortability and LiNGAM, FICO demonstrates the fastest execution, particularly in high-dimensional settings.

## THE USE OF LARGE LANGUAGE MODELS

- **Paper Writing:** LLM was used to polish the text of this paper. We ensure that all modifications were manually verified to be accurate and consistent with the authors' intended meaning.
- **Programming Assistance:** LLM assisted in implementing the QPE-k algorithm (Section 4.2) and generating code for some figures (Figures 1, 3 and 6). All code developed with LLM assistance underwent unit testing and manual debugging. All other experimental procedures and data processing were performed manually.
- **Runtime Environment Fixes:** LLM provided suggestions for resolving issues with older baseline runtime environments in newer versions and assisted in connecting R and Python, allowing all experiments to be conducted within Python.

