# OpenReview forum: "Causal Discovery via Quantile Partial Effect"
_ICLR.cc/2026/Conference — ICLR 2026 Poster_

### Official Review · Reviewer_w7Hq · 2025-10-23

**Soundness:** 3
**Presentation:** 3
**Contribution:** 3
**Rating:** 6
**Confidence:** 2

**Summary:**

The paper proposes Quantile Partial Effect (QPE) as an observational-level object for causal discovery. QPE is defined via conditional quantiles and admits multiple equivalent forms, enabling estimators that do not assume a specific functional causal model (FCM). The core theory shows identifiability of causal direction when each component of the QPE lies in a finite linear span of basis functions, formalized by Theorem 3.6. Building on this, the authors give two bivariate procedures: (i) a kernel QPE with an OLS basis test (QPE-k) and (ii) a flow-based QPE with a neural basis test (QPE-f)—and report strong empirical results. For multivariate discovery, they leverage a PDE link between QPE and the score function to derive Fisher-Information Causal Ordering (FICO), which greedily removes leaf nodes using marginal score variances under an additional assumption (Assumption 4.5). Experiments show QPE-f competitive or superior to many bivariate baselines, and FICO performs competitively on synthetic and real benchmarks.

**Strengths:**

a. The equivalence of QPE and the Wronskian-based identifiability criteria are clear and self-sufficient at the observational level.

b. Unifies causal-velocity style reasoning with quantile/CDF views and score-based analysis.

c. Two practical bivariate pipelines: kernel, OLS basis test (fast), and flow, neural basis test (accurate). And, for multivariate cases, FICO is simple to implement given score estimators.

**Weaknesses:**

a. The central finite-span assumption (Assumption 3.5) is elegant but potentially restrictive; practical guidance on choosing/validating basis sets is limited. How sensitive are conclusions to basis mis-specification?

b. Assumption 4.5 for FICO appears to lack easy a-priori diagnostics.  The performance on Sachs suggests it fails in practice, which narrows applicability.

c. Multivariate ranking depends on accurate scoring function estimation; the paper uses first-order scores for speed but does not deeply analyze robustness to estimator bias or noise.

**Questions:**

a. How is the assumption on lines 96–97 used downstream? e.g., in which lemmas/theorems or in the estimator design. and what would break without it?

b. The paper claims that QPE does not require the (causal) Markov assumption. However, some steps appear to rely on independence of noise terms (exogeneity)?

c. How should practitioners select basis families for Assumption 3.5 in new domains? Could the authors provide data-driven diagnostics or model selection among candidate bases?

d. How sensitive are ODRs to the estimator class and its hyperparameters in FICO?

e. Why Sachs is hard? Can the authors analyze where Assumption 4.5 breaks on Sachs?

---

> ### Author Response · Authors · 2025-11-20
>
> We thank the reviewer for pointing out potential weaknesses of our method in practical applications.
>
> ### Weaknesses
>
> > **(W1)**. Lack of validation for the basis set, and sensitivity to misspecification.
>
> The validation and selection of the basis set will be part of our future research. Regarding sensitivity to misspecification, please see our response in "**Common Questions, Q3**".
>
> > **(W2)**. FICO's assumptions lack prior diagnostic methods, and the results on Sachs indicate limited applicability.
>
> We acknowledge this issue. FICO's assumptions are rather difficult to interpret, but this is a trade-off for broadening the theoretical scope of the method. For the limitations regarding the Sachs dataset results, please see "**Common Questions, Q2**".
>
> > **(W3)**. FICO depends on score function estimation, but the robustness to estimator bias or noise was not analyzed.
>
> For fairness, we used the same score function estimator for all score-based causal discovery methods in our experiments. This, to some extent, reflects the performance of these methods under similar bias or noise. More advanced score function estimators can be applied and analyzed in future work.
>
> ### Questions
> > **(Q1)**. How are the assumptions in lines 96-97 used downstream, and what is the impact of violating them?
>
> The general assumptions in lines 96-97 are used in almost all theorems. These assumptions ensure that partial derivatives up to order $k$ are continuous and that the mixed partial derivatives commute, which legitimizes the derivation of the corresponding conclusions. If these assumptions are violated, these conclusions would be problematic.
>
> > **(Q2)**. QPE is claimed to not require the Markov assumption, but some steps seem to require noise independence?
>
> The definition of QPE and the achievement of identifiability do not require it. However, some steps do indeed involve the Markov assumption (please see "**Common Questions, Q1**").
>
> > **(Q3)**. How is the basis chosen in practice? Is a data-driven or model selection method provided?
>
> As with W1, we acknowledge that the current work is indeed lacking research in this area. Please see "**Common Questions, Q3**".
>
> > **(Q4)**. How sensitive is FICO's performance to the score estimator and hyperparameters?
>
> We used the score estimator from (Rolland et al., 2022), which has only one hyperparameter, $\eta$, set to $0.001$. Theoretically, FICO and other score-based causal discovery methods are sensitive to the score estimator, as their performance depends on the accuracy of the score estimation.
>
> > **(Q5)**. Why did it fail on Sachs? Is there an analysis of Assumption 4.5?
>
> We provide some qualitative explanations in "**Common Questions, Q2**", based on the analysis in Appendix B.2. However, since the underlying model of the Sachs dataset is unknown, it is impossible to precisely calculate the degree to which Assumption 4.5 is violated.

---

> > ### Comment · Reviewer_w7Hq · 2025-11-26
> >
> > Thank the authors for the response. Most of my concerns have been answered.

---

### Official Review · Reviewer_vus6 · 2025-10-27

**Soundness:** 4
**Presentation:** 4
**Contribution:** 2
**Rating:** 6
**Confidence:** 4

**Summary:**

The authors propose estimating the quantile partial effect (QPE) as a means of functional causal discovery. The QPE generalizes causal velocity to the distributional level, which avoids making SCM assumptions. They derive novel conditions for both identifying causal direction in the bivariate case as well as sink nodes in a multivariate setting. Based on these conditions, they propose two methods for bivariate causal discovery (QPE-f, QPE-k) and an order-based approach for the multivariate case (FICO).

**Strengths:**

- The authors propose a Wronskian criterion for bivariate causal discovery that is not only theoretically interesting but also directly informs novel methodology based on a practical test, which shows strong performance in the common benchmarks.
- Mathematical erivation and discussions (especially in the Appendix) are complete and rigourous
- Experiments are exceptionally comprehensive and easy to follow
- Overall writing is clear, the paper is well-structured and logical

**Weaknesses:**

- The theoretical generalization beyond causal velocity is limited in practice, since the discussion in Appendix A.2 on identifiability still requires the SCM assumption which equates QPE and causal velocity.
- Assumption 4.5 for FICO is specifically designed for Corollary 4.3 and is difficult to interpret in practice (although I appreciate the authors acknowledging this).
- The resulting algorithm, FICO, seems very similar to the SCORE algorithm (Rolland et al., 2022).

**Questions:**

- Question on your Appendix A.2: You show that, given a joint distribution (up to $p(x)$) the class of forward models $p(y|x)$ are non-parametric, while the class of backward models $p(x|y)$ are parametric, based on an extension of the ANM argument in Hoyer et al., 2008. My question is this: does the (k+2)-th order ODE for $\log p(x)$ somehow depend on the anti-causal model, either via $\eta_{X \mid Y}$ or via the basis functions $\phi(x)$ in Assumption 3.5? And if it does, will changing these unknown properties also change the ODE?

- Question on FICO: can you clarify the relationship between FICO and SCORE? You still have to estimate the score function for FICO, correct? SCORE looks for the smallest variance in score across nodes, while FICO looks for the smallest FI, or squared score. But asymptotically under mild conditions the FI and variance are the same (the experimental results also look very similar). Is Corollary 4.3 a condition on when we expect SCORE to work for non-linear, non-additive noise models?

---

> ### Author Response · Authors · 2025-11-20
>
> We thank the reviewer for their review and suggestions.
>
> ### Weaknesses
>
> > **W1**. The generalization is limited in practice, as the assumptions used in the results of Appendix A.2 equate QPE and causal velocity.
>
> In short, the use of the same assumptions for QPE as for causal velocity in Appendix A.2 was a trade-off made to quantitatively describe the scarcity of unidentifiable cases. In contrast, the results in the main text do not require monotonicity and Markov properties, as detailed in our response to "**Common Questions, Q1**".
>
> > **W2**. Assumption 4.5 is difficult to interpret.
>
> We acknowledge this issue. It is a trade-off made to broaden the applicability of our method (please see "**Common Questions, Q2**").
>
> > **W2**. The FICO algorithm is somewhat similar to SCORE.
>
> FICO and SCORE are somewhat similar in form, but SCORE's criterion is the variance of the gradient of the score function ($\text{Var}\left[\partial_{x_i}^2 \log p(\boldsymbol{x})\right]$), whereas FICO uses the second moment of the score function ($\mathbb{E}\left[(\partial_{x_i} \log p(\boldsymbol{x}))^2\right]$). In fact, as described in line 451, our criterion is equivalent to the negative expectation of the gradient of the score function used in CaPS (Xu et al., 2024), which is $-\mathbb{E}\left[\partial_{x_i}^2 \log p(\boldsymbol{x})\right]$.
>
> ### Questions
>
> > **Q1**. Do the results in Appendix A.2 depend on the anti-causal model, via $\eta_{X|Y}$ or $\boldsymbol{\phi}$? If so, would changing these terms alter the ODE?
>
> The results in Appendix A.2 do indeed require assuming that $\boldsymbol{\phi}$ is given, but they are always independent of the anti-causal model. This is because, according to lines 886-893, the ODE for $p_X$ is completely determined by the content already given in Assumption 3.5 and Assumption A.1, which can be treated as constants when solving the ODE. Changing $\boldsymbol{\phi}$ does indeed alter the ODE, but it does not change the conclusion about the solution, provided that the Assumption 3.5 still holds for the forward model.
>
> > **Q2**. Could you clarify the relationship between FICO and SCORE? Is Corollary 4.3 a generalization that enables SCORE to work for non-linear non-additive noise models?
>
> Same as our response to W2. To be precise and following your line of thought, Corollary 4.3 is indeed a generalization of **CaPS** (not SCORE), where CaPS only supports additive noise, while Corollary 4.3 is general.

---

> > ### Comment · Reviewer_vus6 · 2025-11-24
> >
> > I'd like to thank the authors for their thoughtful response. I have a better appreciation now for the difference between your proposed Corollary I.1 versus the Corollary A.2 in the paper. Yes, it seems that the causal velocity work and indeed previous work on ANM/LSNM etc. seemingly rely on the Markovian SCM structure, but it's actually realized through distributional quantities, which I believe you've shown to be summarized in the QPE. It may be in the author's interest to highlight in the paper that these causal discovery approaches in fact don't require counterfactual assumptions in general (is this true?).
> >
> > Also, apologies that I was unaware of CaPS. Indeed it seems you are giving a new condition that documents when CaPS works, and making a small computational change. I'm totally fine with this, but I don't see the point of the experiments in Section 5.2 (the results are identical to CaPS), and presenting FICO as a new algorithm with a new name.
> >
> > I remain positive about the paper, in particular about the new Corollary I.1. Personally, I would much prefer the paper to spend more time including and discussing elements of I.1, Appendix A, and B.2 in the main text, rather than presenting FICO as a new method. I will suggest acceptance contingent on some restructuring of how the multivariate contributions are presented:
> >
> > 1) frame FICO as a algorithmic modification and additional theory for CaPS motivated by the QPE interpretation instead of a new method,
> > 2) bring a shortened version of B.2 and maybe the bottom row of Table 10 up to the main text,
> > 3) remove section 5.2 (or just make the point that the modification does not affect CaPS performance in practice).
> >
> > Note I don't think it's necessary to physically make these changes within the discussion period, but if the authors agree for the final version I will raise my score. This will end up being documented on openreview anyway.

---

> > > ### Author Response · Authors · 2025-11-25
> > >
> > > We thank the reviewer again for the suggestions to help us improve the presentation of the manuscript.
> > >
> > > > It is suggested to emphasize that these approaches, in fact, don't require counterfactual assumptions in general.
> > >
> > > We will supplement the discussion following Corollary I.1 with this point.
> > >
> > > > It is suggested to reduce some content related to FICO, describe it as a supplement to CaPS, and devote more space to discussing Corollary I.1, Appendix A, and B.2 in the main text.
> > >
> > > Regarding the presentation of Section 2 in the main text, and specifically that the conclusions in Corollary I.1 and Appendix A should be explicitly featured in the main text, we have already committed to this in our Global Rebuttal.
> > >
> > > Considering the issue of how to present the multivariate contributions, we will make the following modifications in the final version:
> > >
> > > *   Replace the discussion regarding Corollary 4.3 in lines 324-343 with the conclusions from Appendix B.2, and move Corollary 4.4 to Appendix B.2.
> > > *   Clarify the relationship between FICO and CaPS in Section 4.2, describing FICO as a supplement to CaPS in terms of theory and computation.
> > > *   We will revise Section 5.2 to focus on the comparison between FICO and CaPS, while moving comparisons with other methods (such as the full versions of Table 3 and Figure 3) to Appendix D.3.
> > >
> > > Before the end of the rebuttal phase, we will add a "Part IV" to the Global Rebuttal to summarize all specific modifications, rather than directly uploading a revised version. This is because we have used line numbers or indices from the original version to locate content during the rebuttal process. The revised version might cause confusion regarding this and hinder discussions with other reviewers.

---

> > > > ### Comment · Reviewer_vus6 · 2025-11-27
> > > >
> > > > Sounds good! I'm raising my suggestion to "Accept". It will be a nice contribution to the literature.

---

### Official Review · Reviewer_W1w7 · 2025-10-30

**Soundness:** 3
**Presentation:** 2
**Contribution:** 3
**Rating:** 4
**Confidence:** 4

**Summary:**

The paper proposes an approach to causal discovery from observational data, based on the notion of quantile partial effect (QPE). In particular, three distinct approaches utilizing QPE are proposed:

(i) A kernel-based, non-parametric approach for bivariate discovery which assumes that the QPE can be represented as a linear combination of known basis functions,

(ii) A parametric approach for bivariate discovery using the idea of normalizing flow, where each flow is parameterized by a neural network,

(iii) A parametric approach, which recovers a causal ordering among a set of variables, based on the Fisher information matrix.

**Strengths:**

(S1) Causal discovery is an important topic in causal inference, and further advances in this context are needed. The paper deals with an important topic.

(S2) The framing of previous methods (LiNGAM, ANM, HNM) as specific cases of the general setting in which QPE lies in the linear span of fixed basis functions, seems interesting and novel. Getting a generalization of current approaches assuming different functional constraints is a good direction to explore.

(S3) The performance of the flow-based (neural) approach for QPE testing seems to perform quite well empirically.

**Weaknesses:**

(W1) The exposition of the paper could be substantially improved. A range of assumptions are discussed, and in the current version, it is non-trivial to follow for which of the proposed approaches exactly which assumptions are used. This could be streamlined through an overview table.

(W2) Causal Velocity comparison is misleading: Some of the language around causal velocity / bijective causal models seems concerning (abstract, after Proposition 3.3).

When a bijective causal model assumption is invoked, some collapse of the layers does occur. This seems to be an important point for causal velocity.

QPE is defined purely at the observational level, and does not pertain to any structural assumptions — it is simply a functional of the underlying observational distribution. Therefore, comparing these two in such a way (saying that “QPE does not need a monotonicity assumption”) seems quite confusing. It seems more likely that (i) the definition of QPE does not require monotonicity; but (ii) for the proposed discovery methods to work, monotonicity is assumed?

(W3) Removal of the Markovian assumption: the claims on the removal of the Markov assumption are very curious. Most work in causal discovery using functional assumptions considers the Markovian case. If the authors are truly relaxing this assumptions, this seems like a big deal, and should be emphasized strongly. If not, explaining exactly how the Markov assumption is “not needed” is very important — is the case again that QPE can be defined without invoking the Markov assumption, but to apply QPE-based discovery methods, the Markov assumption is needed? If the latter is true, these claims need to be reframed properly.

(W4) It is not uncommon for causal discovery methods to provide some form of formal guarantees that the method will work with high probability etc. Are the formal guarantees needed / missing from the paper?

(W5) Line 289 states that “OLS assumes a Gaussian distribution”. OLS is a method, which does not assume any distribution implicitly. Of course, with Gaussian noise, OLS may exhibit specific properties (become the MLE etc.). Clarifying this would be important.
Furthermore, in this setting arbitrary distribution of $x_t, y_t$ can now be handled — is this previously not the case? This is definitely worth clarifying.
This lack of clarity on the exact assumptions connects back to point (W1).

(W6) In Lines 338, it is said that $\partial_y \psi_{Y | X,i} = 0$ is “e.g. ANM”. To be clear, in this context, additive noise is assumed in addition to the fully parametric model used in this setting?

ANM usually allows arbitrary functions $a(X)$; which is not the case here? It would be more precise to say “additive noise” than ANM in this case?

Same comment for the “e.g. HNM” model.

(W7) Assumption 4.5 seems to be important for the FisherInfo-based approach. I currently have no idea how one might judge the validity of such an assumption, or even start thinking about it? It seems that this is insufficiently discussed, and raises the concern whether the exact condition needed for the method is assumed, regardless of its plausibility.

I note that I would be willing to reconsider my grade in light of the authors' responses to weaknesses / questions.

**Questions:**

(Q1) Is the representation in Table 1 actually true?: specifically, for PNL-ANM and PNL-AHM models, the basis functions $\bar g, g^{-1}\bar g$, etc. are needed. Isn’t it generally the case that the function $g(\cdot)$ is not known in these settings? How does this tie with the fact that the basis functions in $\Phi$ need to be known?

(Q2) Relating to the previous point, is there any nuance w.r.t. choosing the basis? The choice of the kernel bandwidth parameter is mentioned as a limitation, but the choice of basis (which is even more fundamental?) is not really mentioned.

(Q3) The performance of QPE-f seems to be very strong empirically. I wonder, for settings where the data-generating model is actually an ANM, does the ANM method perform better? (similar question for LiNGAM & others). If I understand, current experiments would seem to imply the answer is no? If so, it is curious.

(Q4) Relating to previous point — given the strong performance of QPE-f, is there any way to move beyond bivariate discovery with such an approach?

---

> ### Author Response · Authors · 2025-11-20
>
> We thank the reviewer for pointing out these weaknesses to help us improve the manuscript.
>
> ### Weaknesses
>
> > **(W1)**. An overview is needed to illustrate the dependency relationships between the assumptions and theorems.
>
> We have provided a version in the "Assumptions & Theorems Overview" section of our general response, which will be added to the manuscript.
>
> > **(W2 & W3)**. The definition of QPE does not require the Markov property & monotonicity, but are they required for causal discovery to work?
>
> The use of these two assumptions depends on the level of identifiability guarantee one aims to achieve (see "**Common Questions, Q1**"). Specifically, the main text describes one approach to achieve identifiability by leveraging the "asymmetry in the shape of the observed distribution," an additional assumption that is independent of the Markov and monotonicity properties. On the other hand, Appendix A.2 presents an alternative approach which, despite requiring the Markov and monotonicity properties, provides quantitative guarantees (see W4).
>
> > **(W4)**. Lack of formal guarantees, e.g., with high probability.
>
> Appendix A.2 provides a quantitative guarantee similar to that in (Hoyer et al., 2008). It is not a probability-based description but is based on manifold dimension and measure theory, as detailed in lines 909-915.
>
> > **(W5)**. OLS makes no distributional assumptions and is equivalent to Gaussian noise under MLE; this needs clarification. Additionally, it needs to be clarified whether the distributions of $x_t, y_t$ can be arbitrary.
>
> Indeed, we will clarify this in the revised version for improved rigor. The phrase "$x_t, y_t$ can be arbitrarily distributed" in line 295 mainly refers to the "irregularly distributed samples" mentioned in line 288, in contrast to OLS-based methods that require grid samples. We will also revise the wording on this point for clarity.
>
> > **(W6)**. Is the use of ANM in line 338 imprecise? Since $a(X)$ in ANM is arbitrary. The same applies to HNM.
>
> The use of "ANM" in line 338, where $\partial_y \psi_{Y | X,i} = 0$, is indeed accurate. According to Table 1, the QPE for an ANM is $\psi_{Y | X}=\nabla a$, and since $a$ is independent of $y$, it follows that $\partial_y \psi_{Y | X,i}=0$. Similarly, the description for HNM is also accurate.
>
> > **(W7)**. The validity of and discussion on Assumption 4.5 are insufficient.
>
> We acknowledge this issue. It is a trade-off made to broaden the applicability of our method (see "**Common Questions, Q2**"). Specifically, we attempted to further discuss Assumption 4.5 in Corollary 4.4, the entirety of Appendix B.2, and the experiments in Appendix D.3 on page 28. However, for now, we are limited to a qualitative analysis.
>
> ### Questions
>
> > **(Q1)**. Is Table 1 correct? For the PNL-ANM model, the basis functions are related to $g$. how does this reconcile with the assumption that the basis is known?
>
> Assumption 3.5 cannot be directly applied to post-nonlinear models (where $g$ is unknown), it requires $g$ to be known. We will clarify this in the revised version. For example, in PNL-ANM, if $g$ is a tanh or sigmoid function, the basis set would be quadratic.
>
> > **(Q2)**. How to choose the basis? This seems unmentioned.
>
> We mentioned this problem in the conclusion as a direction for future work. Currently, there is no particularly good solution, and we can only explore it through heuristics or empirical methods (see "**Common Questions, Q3**").
>
> > **(Q3)**. Do ANM-based methods perform better on ANM-generated data? The experiments seem to suggest this is not the case?
>
> Theoretically, since both the QPE method and ANM-based methods support the ANM dataset, they should perform comparably. In practice, this is indeed reflected on ideal synthetic datasets like AN, AN-s, LS, LS-s, and MNU (for instance, CAM also performs reasonably well). The reason why experimental results on some datasets seem to suggest otherwise is either that the data was not generated from an ANM, or that components that are harder to distinguish were intentionally introduced (e.g., the SIM series), making purely ANM-based methods empirically less robust to these situations.
>
> > **(Q4)**. Can QPE-f be extended to the multivariate case?
>
> Currently, this remains an open research question, as described in the conclusion. The main challenge stems from QPE estimation. We have attempted to extend QPE-f directly to the multivariate case, but experimental results show that the curse of dimensionality is very apparent in QPE estimation, causing the estimated QPE to deviate significantly from the true QPE.

---

> > ### Comment · Reviewer_W1w7 · 2025-11-22
> >
> > I thank the authors for the detailed response. As a follow-up, I maintain the suggestion to refine the part comparing causal velocity and QPE -- having re-read this part, I think this could easily confound the reader, since causal velocity / QPE seem to be fundamentally different objects in terms of how they are defined and the roles they play.
> >
> >  The biggest concern that remains is related to Assumption 4.5. When you write
> >
> > > Experiments also show this assumption holds in most synthetic datasets.
> >
> > what is this statement based on? Are you basing this statement on good empirical performance on synthetic datasets? One would think that this assumption is not directly testable.
> >
> > If so, it seems that acknowledging this limitation more explicitly by stating that: (i) assumption is not testable; (ii) difficult to interpret currently, would improve the transparency of the paper's claims.

---

> > > ### Author Response · Authors · 2025-11-22
> > >
> > > We further thank the reviewer for providing feedback, especially regarding the weaknesses in our rebuttal.
> > >
> > > > **1**. Improve the comparison between QPE and causal velocity, as their definitions and roles are completely different.
> > >
> > > We will supplement and adjust lines 154-158. Specifically, our original statement, "QPE does not require monotonic mechanisms or the Markov assumption," will be adjusted to "**the definition** of QPE does not...". Additionally, we will add that "achieving cause-effect identifiability through QPE also does not require these assumptions" to pave the way for the subsequent discussion and help readers follow our line of reasoning.
> > >
> > > > **2**. Issues with the statement "Experiments also show this assumption holds in most synthetic datasets" in the "Common Questions and Responses" section, and a request to improve the transparency of claims about Assumption 4.5.
> > >
> > > We acknowledge that this statement lacks precision. Our intended meaning was, as stated in line 1532, "This empirically suggests that the underlying assumptions or characteristics of these methods are implicitly satisfied." Indeed, this is only an empirical conclusion, and we have corrected this point in our Global Rebuttal.
> > >
> > > Regarding the issues of testability and interpretability, this will be emphasized in line 342, where we acknowledged the weakness in interpretability. In the revised version, we will explicitly acknowledge the weakness regarding testability and explain that these weaknesses are a trade-off for generalization, as stated in "Common Questions, Q2".

---

> > > > ### Comment · Reviewer_W1w7 · 2025-11-22
> > > >
> > > > Thanks for the response. As already said, I think the paper has some good ingredients, and I have adjusted the score accordingly.

---

### Official Review · Reviewer_mFRa · 2025-11-02

**Soundness:** 4
**Presentation:** 3
**Contribution:** 4
**Rating:** 6
**Confidence:** 4

**Summary:**

The paper studies causal discovery from observational data using distribution shape constraints. Authors motivate non-graphical assumptions about the gradient of the joint distribution that emphasize the asymmetry between the cause and effect in a bivariate case, and extend the results to multivariate causal discovery by iteratively finding the last variable in the causal order. Through examples, they show how the shape constraints introduced in their work is distinct from the graphical constraints commonly employed in this problem, and experiments validate usefulness of their approach.

**Strengths:**

1. Clear statement of the assumptions underlying the theoretical results.
2. Motivating examples throughout the paper allow the more curious reader to understand the motivation.
3. Algorithm 1 is impressively simple, and the connection between fisher information and the shape constraints are quite insightful.

**Weaknesses:**

1. Appendix A3 contains very important remarks that deserve to be presented in the main text. The paper argues that the shape assumptions are more *relaxed* than the existing assumptions employed in causal discovery, and this subtle points needs to be established carefully, or else the reader may find the contributions not well-motivated.
2. Sections 3.3 and 3.4 are not contributing to what comes later in the manuscript; they are useful extensions of 3.2, though, I recommend a restructuring as they hurt the coherency.
3. Missing discussion about related work [1,2,3,4] that would enrich the work.

[1] Hyvarinen, A., Sasaki, H., & Turner, R. (2019, April). Nonlinear ICA using auxiliary variables and generalized contrastive learning. In The 22nd international conference on artificial intelligence and statistics (pp. 859-868). PMLR.

[2] Jalaldoust, K., Salehkaleybar, S., & Kiyavash, N. (2025). Multi-domain causal discovery in bijective causal models. In Proceedings of the Fourth Conference on Causal Learning and Reasoning.


[3] Guo, S., Tóth, V., Schölkopf, B., & Huszár, F. (2023, December). Causal de finetti: On the identification of invariant causal structure in exchangeable data. In Thirty-seventh Conference on Neural Information Processing Systems.

[4] Reizinger, P., Sharma, Y., Bethge, M., Schölkopf, B., Huszár, F., & Brendel, W. (2023). Jacobian-based causal discovery with nonlinear ICA. Transactions on Machine Learning Research.

**Questions:**

Please address my concerns above.

---

> ### Author Response · Authors · 2025-11-20
>
> We thank the reviewer for pointing out issues regarding presentation and missing literature.
>
> ### Weaknesses
>
> > **W1**: The content in Appendix A3 is important and should be presented in the main text.
>
> We thank the reviewer for this suggestion. To address the concerns raised by multiple reviewers about our assumptions (see "Common Questions, Q1"), we will move more relevant content into the main text to make our presentation and motivation clearer.
>
> > **W2**: Sections 3.3 and 3.4 lack coherence with the preceding sections.
>
> We will adopt this suggestion and move these sections as subsections into a new section focused on Bivariate Causal Discovery. This will improve the paper's coherence.
>
> > **W3**: Some literature is missing.
>
> We will add the mentioned literature in the revised version.

---

### Author Response · Authors · 2025-11-20
**Gloabl Author Rebuttal (Part I)**

# **Common Questions and Responses**

> **Q1. Although you claim that QPE does not require monotonicity and Markov properties, some steps still use these two properties (`W1w7:W2&3`; `vus6:W1`; `w7Hq:Q2`).**

Based on the dependency chain of our theorems, it is true that Proposition 3.3 and Corollary A.2 utilize monotonicity and Markov properties. We clarify their specific roles below, and emphasize that this does not contradict our main claim that *observational distributions alone are sufficient for cause-effect identifiability*.

*   **Use in Proposition 3.3:** The use of monotonicity and Markov properties in Proposition 3.3 is to ensure counterfactual identifiability. Crucially, this is a requirement for the definition of *causal velocity*, not for the definition of QPE itself.

*   **Use in Corollary A.2:** Corollary A.2, one of the core guarantees for cause-effect identifiability, does use these properties. However, it serves as a supplement to provide a *quantitative guarantee*, which was missing from the discussion in lines 212-215 of the main text.

*   **Implicit Properties in QPE-f:** This is an inherent design of the causal flow model. Although monotonicity and Markov properties are implied, the flow in QPE-f is merely a tool to estimate QPE. According to Proposition 3.3, the estimated QPE is always consistent with the true QPE, and the true underlying model does **not** require monotonicity or Markov properties (see Appendix A.3 for some examples).

In fact, the core argument for why our causal discovery works is presented in lines 212-215, albeit informally through examples. In the following thread, we provide a formalization of lines 212-215 (Corollary I.1). This corollary guarantees cause-effect identifiability through an additional assumption, "**Asymmetry in the Shape of the Observational Distribution**" (Assumption I.4, mentioned in the abstract, introduction, and lines 212-215), and it works **without** requiring monotonicity or Markov properties. This directly supports our claims in the abstract, introduction, and TL;DR, and is one of our core theoretical contributions.

In summary, Corollary I.1 (in the main text) and Corollary A.2 (in the appendix) are complementary. Both prove cause-effect identifiability. The former does not need monotonicity/Markov properties but lacks quantitative guarantees (we provide some examples in Appendix A3 to compensate). The latter requires these assumptions but provides quantitative guarantees. Given the significant confusion this has caused, **we will replace lines 212-215 with the formal statements of Assumption I.4 and Corollary I.1 in the revised version.**

> **Q2. On the interpretability and practical issues of Assumption 4.5 (`W1w7:W7`; `W1w7:W2`; `w7Hq:Q5`).**

Regarding interpretability, we attempted to analyze Corollary 4.4 (a special case of Corollary 4.3 for HNM) and the entire Appendix B.2 (which introduces a Gaussian noise assumption). However, we found its interpretation remains difficult and largely qualitative (e.g., we added an empirical validation of Appendix B.2 in Appendix D.3, page 28). This is a trade-off: Assumption 4.5 allows Corollary 4.3 to work directly in **general non-linear, non-additive settings** (addressing `vus6:Q2`), thus broadening the theoretical and practical scope of FICO. Experiments also empirically suggests that the underlying assumptions or characteristics of these methods are implicitly satisfied in most synthetic datasets. However, Assumption 4.5 has inherent interpretation and testing difficulties. For instance, even under the ANM assumption, it requires a rather complex condition (CaPS; Xu et al., 2024), although this condition can be formally expressed rather than only qualitatively described.

Regarding practical use, our method underperforms on the Sachs dataset. Our results show that all score-based methods perform poorly on Sachs, suggesting that their **common underlying assumptions may be violated** in this case. We can only offer a qualitative analysis here, based on the conclusions in Appendix B.2. In short, we found that for the standardized Sachs distribution, the variance of the effect w.r.t. the cause changes too drastically. This violates the ideal conditions in lines 1103-1108, making it highly likely that Assumption 4.5 is not satisfied.

---

> ### Author Response · Authors · 2025-11-20
> **Gloabl Author Rebuttal (Part II)**
>
> > **Q3. How to choose the basis and what is the impact of basis misspecification (`W1w7:Q2`; `w7Hq:W1&Q3`).**
>
> Theoretically, the basis is an inductive bias that generalizes past methods like ANM, HNM, and PNL, as shown in Table 1, where each model corresponds to a special case of basis. The QPE method makes this inductive bias explicit, offering a more flexible selection space. Our theory guarantees cause-effect identifiability as long as the basis is **finite and fixed** (Assumption 3.5).
>
> Practically, the choice of basis requires consideration. The final basis choice for our QPE-f method is detailed in Appendix D.1. We compared three types: `Hyper` (fully flexible), `Lowrank` (finite), and `Poly` (fixed, differing only in degree), where only `Poly` fully satisfies Assumption 3.5. The results in Table 6 show that **`Poly` is empirically optimal on most datasets**, which indirectly highlights the importance of Assumption 3.5. However, how to select the optimal basis function, especially without a validation set or labels, is a valuable direction for future research.
>
> Furthermore, misspecifying the basis does impact performance, as clearly shown in Table 4. Since the SIM dataset is generated from an ANM, nearly all QPE-f hyperparameter settings achieved the best empirical performance only when $K=0$ for the `Poly` basis. Similar trends are seen in other synthetic datasets in Table 6 (e.g., all SIM-type datasets were empirically optimal at $K=0$). To be transparent, and speaking quantitatively, misspecifying the basis can cause significant performance degradation. For example, in Table 4, the accuracy difference for QPE-f between $K=0$ and $K=2$ under the `Poly` basis can be as high as 10% for some hyperparameter settings.
>
> However, we must emphasize that while the basis selection remains an open question, our work **significantly extends the theoretical scope for cause-effect identifiability** compared to FCM methods like ANM and HNM. As special cases of our theory, these methods also implicitly assume a specific basis (as shown in Table 1) and are thus *potentially affected* by basis misspecification, but they lack the flexibility of our explicit formulation.
>
> ---
>
> # **Assumptions & Theorems Overview**
>
> To clearly describe our work and address the concerns raised in `w1w7:W1`, we provide a comprehensive summary of all theoretical results presented in our paper. This summary includes both explicitly stated theorems and those implicitly mentioned in the context, along with a detailed map of their dependencies.
>
> ## **Implicitly Mentioned Theoretical Results**
>
> Below, we summarize several theoretical points that were mentioned in the context but not formally stated as propositions, primarily due to space constraints. We denote these with the prefix 'I' (for 'implicit'). In the revised manuscript, we will consider formally stating the most crucial of these (particularly those that might cause confusion for readers, such as the discussed Corollary I.1) in either the main text or the appendix to provide emphasis and clarity. All other theoretical results discussed hereafter are explicitly stated in the paper.
>
> - **Assumption I.1 (Global Assumptions):** Absolute positivity, continuity, and $C^k$ smoothness.
>
>   These assumptions are declared in lines 91-97. They enable the derivation of higher-order ODEs/PDEs, as detailed in our response to `w7Hq:Q1`.
>
> - **Assumption I.2 (Markov Property):**
>
>   Mentioned in lines 103-104.
>
> - **Assumption I.3 (Monotonicity):**
>
>   Mentioned in lines 147-148.
>
> - **Assumption I.4 (Asymmetry in the Shape of the Observational Distribution):** For any $Z\in\boldsymbol{X}\setminus Y$ and any $X_i\in\boldsymbol{X}\setminus Z$, we have $W_Z(s_{X_i,Z},\boldsymbol{\eta}_{Z\mid\boldsymbol{X}})\ne 0$.
>
>   This is textually described in lines 212-215. It is crucial to emphasize that when Assumption 3.5 holds, Assumption I.4 depends *solely* on the observational joint distribution (specifically, $\log p_{\boldsymbol{X},Y}$), without requiring the Markov property or monotonicity. This is because $s_{Z,Y}$ is a high-order partial derivative of $\log p_{\boldsymbol{X},Y}$, and $\boldsymbol{\eta}\_{Y\mid\boldsymbol{X}}$ is related to the high-order partial derivatives of $\log p_{\boldsymbol{X},Y}$ and $\boldsymbol{\phi}$, with the latter being given by Assumption 3.5.
>
> - **Lemma I.1 (Assumption I.2 + Assumption I.3 => Counterfactual Identifiability => Causal Velocity is Identified):**
>
>   Mentioned in lines 149-151, with its proof stemming from (Nasr-Esfahany et al., 2023), (Chen & Du, 2025), and only for (Xi et al. 2025).

---

> ### Author Response · Authors · 2025-11-20
> **Gloabl Author Rebuttal (Part III)**
>
> - **Corollary I.1 (Assumption I.4 + Theorem 3.6 => Effect Variable is Identifiable):** If Assumption I.4 and Assumption 3.5 hold simultaneously, then $Y$ is the effect variable.
>
>   This is implied in lines 212-215.
>
>   **Proof:** Since Theorem 3.6 establishes that a necessary condition for Assumption 3.5 ($\psi_{Y\mid\boldsymbol{X},i}\in\text{span}(\boldsymbol{\phi})$) to hold is $W_Z(s_{X_i,Y},\boldsymbol{\eta}\_{Y\mid\boldsymbol{X}}) = 0$, Assumption I.4 implies that $\psi_{Z\mid\boldsymbol{X},i}\notin\text{span}(\boldsymbol{\phi})$. Consequently, Assumption 3.5 does not hold for any $Z\in\boldsymbol{X}\setminus Y$, leaving $Y$ as the only possible effect variable.
>
> ---
>
> ## **Dependency Chain of Theoretical Results**
>
> Here, we illustrate the complete dependency chain among these theoretical results using the format "A+B=>C" to denote that C depends on both A and B. For instance, if C requires assumption A and utilizes the conclusion of lemma B, which in turn depends on assumption D, we represent this as "A+B=>C" and "D=>B". In this case, C ultimately depends on both A and D. We will add a figure illustrating these dependencies to the revised manuscript.
>
> ### **Fundamental Theory of QPE for Causal Discovery (Section 3)**
>
> - **Assumption I.1 => Proposition 3.2**
> - **Assumption I.1 + Lemma I.1 => Proposition 3.3**
> - **Proposition 3.2 => Lemma 3.4**
> - **Assumption 3.5 + Lemma 3.4 => Theorem 3.6**
>
>   We acknowledge that the title of Theorem 3.6 ("Identifiability of QPE in Finite Linear Span") may be somewhat ambiguous. Its primary purpose is to convey that under Assumption 3.5, the QPE is governed by a PDE related to a Wronskian determinant that depends only on the observational distribution. Under certain conditions, this structure emerges *only if* Assumption 3.5 holds. While this does not directly state cause-effect identifiability, it forms the necessary foundation for Corollary I.1.
>
> - **Assumption I.4 + Theorem 3.6 => Corollary I.1**
>
>   Following the dependency chain, Corollary I.1 requires Assumptions I.1, I.4, and 3.5. Notably, the definition of QPE and these three assumptions do not require the Markov property or monotonicity. This is precisely why we claim that the cause-effect relationship (i.e., identifying the effect variable $Y$ in Corollary I.1) can be identified from the observational distribution alone.
>
>   Corollary I.1 successfully eliminates counterfactual-related assumptions like the Markov property and monotonicity, which were required in the identifiability theory of (Xi et al. 2025). However, as noted in `w1w7:W4`, Corollary I.1 does not quantify the "probability" that its formal guarantee holds. Specifically, how "likely" it is for Assumption I.4 to be satisfied. Indeed, without additional constraints, it is difficult to quantify this likelihood. This limitation motivated us to introduce Corollary A.2 in Appendix A.2.
>
> ### **Bivariate Case (Appendix A.2)**
>
> - **Assumption I.2 + Assumption I.3 + "only $p_X$ is unknown" => Assumption A.1**
>
>   Assumption A.1 follows the setup of the bivariate SCM in (Hoyer et al., 2008), which indeed involves monotonicity and the Markov property.
>
> - **Assumption A.1 + Theorem 3.6 => Corollary A.2**
>
>   Corollary A.2 offers an alternative perspective on cause-effect identifiability, addressing the shortcoming of Corollary I.1 by quantifying the "probability" of the formal guarantee. It differs from the implicit Corollary I.1 by replacing Assumption I.4 with Assumption A.1. While Assumption A.1 reintroduces the Markov and monotonicity properties, it allows us to quantify the rareness of non-identifiable cases under this assumption, as shown in (Hoyer et al., 2008).
>
>   Therefore, Corollary I.1 and Corollary A.2 each have their own pros and cons: the former does not require Markov/monotonicity assumptions but lacks a quantitative guarantee, while the latter requires them but provides such a guarantee.
>
> ### **Multivariate Case (Section 4)**
>
> - **Assumption 4.1 + Lemma 3.4 => Theorem 4.2**
>
> - **Theorem 4.2 => Corollary 4.3**
>
>   The inequality relationship provided in Corollary 4.3 enables us to identify the effect variable from a set of variables.
>
> - **Assumption 4.5 => FICO**
>
>   Assumption 4.5 is an instantiation of the conditions required by Corollary 4.3, which allows the FICO algorithm to work directly.
>
> - **Corollary 4.3 + "$\partial_y^2 \psi_{Y\mid \boldsymbol{X}, i}=0$" => Corollary 4.4**
>
>   Corollary 4.4 is our attempt to provide a specific interpretation of Corollary 4.3 under the condition $\partial_y^2 \psi_{Y\mid \boldsymbol{X}, i}=0$ (e.g., in an HNM). However, the applicability of FICO is not limited to the conditions of Corollary 4.4.

---

> > ### Author Response · Authors · 2025-12-02
> > **Gloabl Author Rebuttal (Part IV)**
> >
> > We would like to thank the reviewers for their engagement and for their valuable suggestions, which have helped improve the manuscript.
> >
> > We have uploaded the revised version, in which modifications are highlighted in blue and additions in orange. Furthermore, `todonotes` in the margin indicate the specific reviewer comments or common questions addressed by each change.

---

### Meta-Review · Area_Chair_NPFV · 2026-01-05

**Summary:**

This paper proposes an approach for causal discovery from observational data based on quantile partial effect. The reviewers generally found the work to be novel and provide new insights for causal discovery, framing some of the previous functional causal models as specific cases. While there are initial concerns e.g. about the need for Markov assumption, challenge in selecting basis, interpretation of certain assumptions, most of them were addressed during the rebuttals, and I thereby recommend acceptance.

**Reviewer Concerns:**

Reviewer concerns addressed by the rebuttal:
- Clarity: Restructuring the paper (moving certain sections around) and adding an overview table.
- Choosing the basis and sensitivity to mis-specification.
- Whether the Markov assumption is not needed (as claimed).
- Connection with SCORE algorithm.

Outstanding concerns:
- The procedure relies on accurate scoring function estimation, and no robustness analysis to estimator bias or noise is provided. (The authors state this as future work.)
- Reviewers raised concern about the assumptions, e.g., judging the validity of Assumption 4.5 and also interpreting it in practice, as evidenced by the under-performance on the Sachs dataset.

**Reviewer Scores:**

Reviewers W1w7 and vus6 decided to increase their score to 6 and 8 respectively, although they are not reflected on the reviews.

---

### Decision · Program_Chairs · 2026-01-26

Accept (Poster)